# Rapid incidence estimation from SARS-CoV-2 genomes reveals decreased case detection in Europe during summer 2020

Maureen Rebecca Smith[1,2,5]✉, Maria Trofimova[1,2,5], Ariane Weber[3], Yannick Duport[1,2], Denise Kühnert [3,4] & Max von Kleist [1,2,4]✉

By October 2021, 230 million SARS-CoV-2 diagnoses have been reported. Yet, a considerable proportion of cases remains undetected. Here, we propose GInPipe, a method that rapidly reconstructs SARS-CoV-2 incidence profiles solely from publicly available, time-stamped viral genomes. We validate GInPipe against simulated outbreaks and elaborate phylodynamic analyses. Using available sequence data, we reconstruct incidence histories for Denmark, Scotland, Switzerland, and Victoria (Australia) and demonstrate, how to use the method to investigate the effects of changing testing policies on case ascertainment. Specifically, we find that under-reporting was highest during summer 2020 in Europe, coinciding with more liberal testing policies at times of low testing capacities. Due to the increased use of real-time sequencing, it is envisaged that GInPipe can complement established surveillance tools to monitor the SARS-CoV-2 pandemic. In post-pandemic times, when diagnostic efforts are decreasing, GInPipe may facilitate the detection of hidden infection dynamics.

[1] Systems Medicine of Infectious Disease (P5), Robert Koch Institute, Berlin, Germany. [2] Bioinformatics (MF1), Robert Koch Institute, Berlin, Germany. [3] Transmission, Infection, Diversification and Evolution Group, Max-Planck Institute for the Science of Human History, Jena, Germany. [4] German COVID Omics Initiative (deCOI), Bonn, Germany. [5] These authors contributed equally: Maureen Rebecca Smith, Maria Trofimova. ✉email: smithm@rki.de; kleistm@rki.de

As of August 2021, the global SARS-CoV-2 pandemic is still ongoing in most parts of the world, with 205 million reported cases worldwide. Novel vaccines of high efficacy have been developed within a year of the outbreak[1,2]. At the time of writing, ~30% of the worlds population had already received at least one vaccination and 15.8% is fully vaccinated. However, the distribution of vaccines is uneven and achieving global herd immunity may pose an extremely difficult, long-term task[3,4]. At the same time, novel variants of concern (VOC) have emerged in high prevalence regions[5,6], which may be able to reinfect individuals[7,8] and escape vaccine-elicited immune responses[9–11]. For example, Manaus, Brazil, witnessed a massive second wave of infections[12], despite the fact that ~80% had already experienced an infection at the onset of the second wave[5].

Because of the evolutionary versatility of SARS-CoV-2 and difficulties in global vaccine distribution, some experts expect that the virus may not be eliminated globally[13]. Even without adaptation to vaccines in the future, it has been postulated that SARS-CoV-2 may resurge[14,15] and surveillance may have to be maintained into the mid 2020s to monitor virus spread and evolution[14].

Currently, the gold standard of SARS-CoV-2 surveillance is diagnostic testing via polymerase chain reaction (PCR) or antigen-based rapid diagnostic testing (RDT). Diagnostic test results currently define infection case reports, which are used to survey epidemiological dynamics and to define thresholds for travel bans and non-pharmaceutical measures. Inevitably, case reporting data are affected by test coverage, which changes when testing policies are adapted. While RDT enables point-of-care diagnosis and is less costly than PCR testing[16,17], gathering and reporting of test results still requires a sophisticated infrastructure, which is difficult to establish and maintain in many developing countries[18]. Independent and complimentary sources of information, such as social media reports[19,20] or waste water analysis[21,22] have been used early on to complement our knowledge of the pandemic dynamics. In addition, many regions of the world sequence SARS-CoV-2 genomes to track virus evolution and the emergence of VOC. The gathered viral sequences are regularly provided to public databases, such as GISAID[23,24]. The genetic data readily holds information about the pandemic trajectory. In this work, we take advantage of the fact that the speed at which SARS-CoV-2 evolves on the population level contains information about the number of individuals who are actively infected.

In the vast majority of cases, SARS-CoV-2 is transmitted within a very short period, only days after infection[25,26]. The consequence is a well-defined duration of intra-patient evolutionary time before transmission. Thus, the number of actively infected individuals is correlated to the rate of divergence of the viral population, implicating an evolutionary signal.

In this article, we introduce the computational pipeline GInPipe, which uses time-stamped sequencing data alone, extracts the evolutionary signal and reconstructs SARS-CoV-2 incidence histories. The approach is inspired by recent work by Khatri and Burt[27], who derived a simple function for the estimation of the current effective population size. Herein, due to the short window of transmission, we anticipate that the effective population size may strongly correlate with the incidence of SARS-CoV-2. We adapt the function derived in ref. [27] and embed it into an automatic computational pipeline (GInPipe) that reconstructs the time course of an incidence correlate $\phi$ merely from SARS-CoV-2 genetic data. GInPipe is validated threefold and performs robustly: (i) against in silico generated outbreak data, (ii) against phylodynamic analyses and (iii) in comparison with case reporting data. We applied the method to SARS-CoV-2 sequencing data from Denmark, Scotland, Switzerland, and the Australian state Victoria to reconstruct their respective incidence histories. Lastly, we utilize the inferred epidemic trajectories to compute changes in the probability that an infected individual is reported and highlight how this probability is affected by changes in testing policies.

## Results

**Incidence reconstruction**. An outline of GInPipe for SARS-CoV-2 incidence reconstruction is shown in Fig. 1a–c. After compiling a set of time-stamped, full-length SARS-CoV-2 genomes, the sequences are assigned to consecutive subsets according to their sampling dates (temporal bins) (Fig. 1a). For each temporal bin $b$, we compute the number of sequences different from a reference (mutant sequences $m_b$), as well as the number of unique sequences (haplotypes $h_b$). These two inputs are used to infer the incidence correlate $\phi_b$ (Fig. 1b). The $\phi_b$ point estimates are smoothed to derive a reconstructed incidence history along the time axis (Fig. 1c). The reconstructed incidence correlates can then be used as a basis to estimate the effective reproduction number $R_e$, as well as the relative case detection rate as outlined below.

**Method validation: in silico experiment**. To confirm that GInPipe is able to reconstruct incidence histories, we performed an in silico experiment. We considered a population of $N(t)$ infected individuals at time $t$ that stochastically generate $N(t+1)$ infected individuals in the next time step $t+1$. Each individual is associated with a virus sequence, which can mutate randomly. Individuals can be removed (the associated sequence is removed), or they transmit their virus (the associated virus is copied over). We record the number of infected individuals per generation, as well as all sequences of the currently circulating viruses. We then use the simulated viral sequences to infer $\phi(t)$ and reconstruct the incidence history, as presented in Fig. 1d, e.

In Fig. 1d, we compare one trajectory of simulated population sizes with the reconstructed incidence histories. The simulated outbreak (red line, right axis) consists of two waves of increasing magnitude. GInPipe robustly reconstructs these dynamics (blue lines and dots, left axis), although the incidence correlates $\phi(t)$ is on a different scale, implying a linear correlation to the number of infected individuals. To assess this correlation, we performed 10 stochastic simulations and compared the $\phi(t)$ point estimates with the corresponding number of infected individuals (Fig. 1e). We observed a strong (Pearson correlation coefficient of $r = 0.98$) and highly significant ($p < 10^{-16}$) linear relationship between the number of infected individuals $N(t)$ and the method's incidence correlate $\phi(t)$.

GInPipe also allows to infer the effective reproduction number $R_e$ from the incidence correlates $\phi(t)$ (details in the "Methods" section). To further assess the accuracy of GInPipe, we compare the $R_e^\phi$ values inferred with the smoothed $\phi$ estimates versus $R_e^{\text{true}}$ values calculated from the simulated pandemic $N_{\text{true}}$. Figure 1f shows the identity plot for $\log(R_e^{\text{true}})$ vs. $\log(R_e^\phi)$, with the respective proportion of qualitatively agreeing or disagreeing predictions in the four quadrants: The top right and bottom left quadrant represents the true positive (TP) and true negative (TN) estimates, and the top left and bottom right quadrants show the false positive (FP) and false negative (FN) estimates respectively. The qualitative accuracy of GInPipe based on the $R_e$ values was calculated as $\frac{\text{TP+TN}}{\text{TP+TN+FP+FN}}$, yielding a value of 0.92. In terms of quantitative agreement of the $R_e$ estimates, the coefficient of determination was $R^2 = 0.77$.

While these simulations represent idealized scenarios, in Supplementary Note 1 we thoroughly evaluated the robustness of GInPipe to incomplete, and sparse data sets, unbalanced and temporally changing sampling rates, to the introduction of unrelated sequence variants, its ability to reconstruct non-smooth

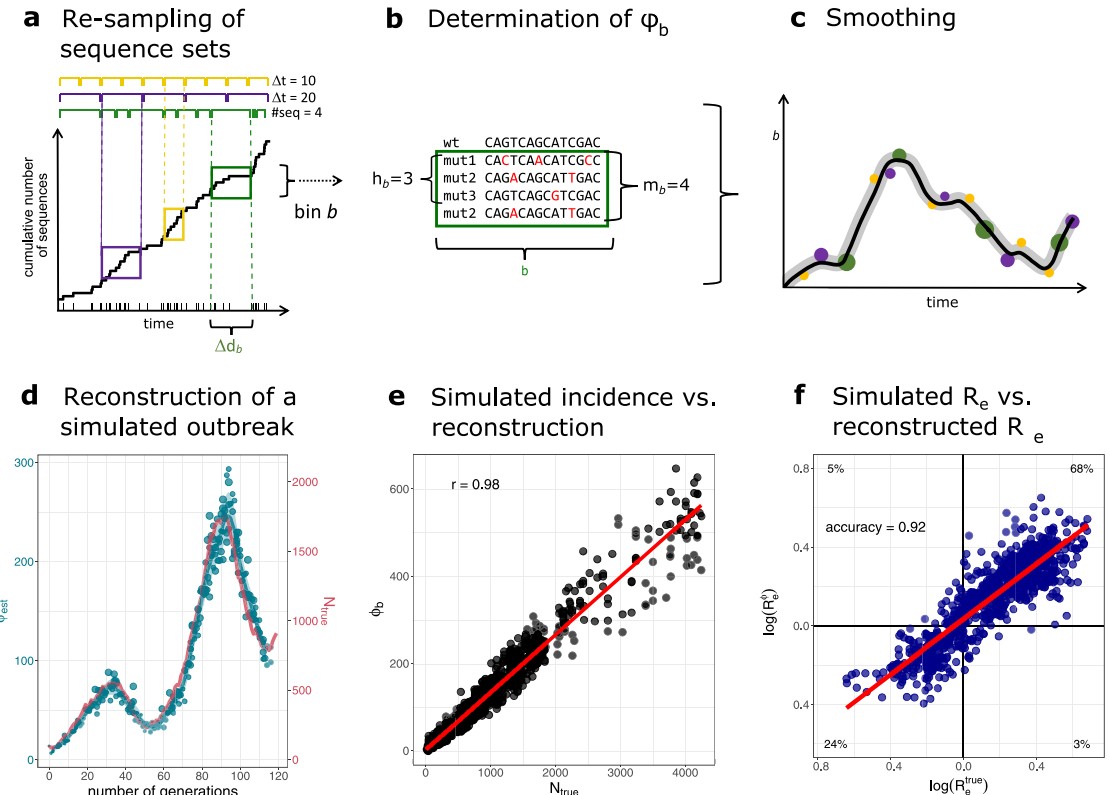

**Fig. 1 Reconstruction of incidence histories using the proposed method. a–c** Schematic of the incidence reconstruction method. **a** The sequences are chronologically ordered by collection date. The line shows the cumulative sum of sequences over time. The sequences are allocated into temporal bins, spanning either the same time frame $\Delta d_b$ (yellow and purple bins) or containing the same amount of sequences (green bins). **b** For each bin, the number of distinct variants $h_b$, as well as the total amount of mutant sequences $m_b$ are used to infer the incidence correlate $\phi_b$. **c** The point estimates for all bins $\phi_b$ (dots) are smoothed with a convolution filter. For uncertainty estimation, the point estimates are sub-sampled and interpolated and 95% confidence intervals highlighted. **d–f** Reconstruction of a simulated outbreak with GInPipe. **d** $\phi$ estimates resemble the underlying population dynamics over time. The blue line shows the smoothed median of the sub-sampled $\phi$ estimates (dots) for a simulated outbreak. The red line indicates true incidence per generation. **e** Dotplot showing the true outbreak size from the simulation $N_{\text{true}}$ versus the $\phi_b$ point estimates for 10 stochastic simulations. The red line depicts the linear fit. **f** The reconstructed incidence correlates $\phi$ allow the determination of the effective reproductions rate $R_e$ as described in the Methods section. The dotplot shows $R_e^{\text{true}}$ (inferred from the true population size $N_{\text{true}}$) versus $R_e^{\phi}$ (constructed with correlate $\phi$) on log scale for the 10 stochastic simulations. The red line shows the linear fit. The proportion of points in each quadrant is shown in the respective partition.

pandemic dynamics, as well as its sensitivity to changes in the pathogen mutation rate and selective pressure.

Our analyses showed, that the method can still reliably reconstruct incidence histories over time when data are missing, or when the sampling rate changes over time. In scenarios of extreme under-sampling, the $\phi$ point estimates have the tendency to yield lower values. However, through the smoothing step the reconstructed incidence trajectories still follow the overall population dynamics (Supplementary Note 1, section SN.1.7). If the sampling changes the evolutionary signal, for example by sampling sequences based on their similarity (and hence lowering the signal), the incidence correlates tend to decrease (Supplementary Note 1, section SN.1.9). If the sampling strategy does not change over the course of the pandemic, GInPipe can still reconstruct the overall population dynamic. However, with altering sampling strategies that perturb the evolutionary signal, difficulties with incidence reconstruction may arise. Therefore, as with, e.g., phylodynamic methods, a consistent strategy of deducing representative samples is believed to ensure GInPipe's performance. We found that selective pressure has no effect on the incidence reconstruction with GInPipe (Supplementary Note 1, section SN.1.14).

If mutation rates become too low, which may be the case for other respiratory infections, and hence not enough signal is given

in the data, GInPipe becomes less accurate, but the incidence can still be reconstructed at the cost of time-resolution (Supplementary Note 1, section SN.1.15).

Finally, we evaluated whether introductions of foreign sequences affect the reconstruction of incidence histories. Even for extreme and unrealistic cases, a stable reconstruction of the underlying dynamic is possible. Yet, a tendency of overestimation can be observed if the introduced sequences constitute more than 10% of the data set and if they do not continue to contribute to the pandemic after their introduction (Supplementary Note 1, section SN.1.12).

**Method validation: phylodynamics.** Phylodynamic methods combine phylogeny reconstruction with epidemic models. For example, the piecewise constant birth-death sampling process (BDSKY)[28] implemented in BEAST2[29], allows the reconstruction of the effective reproduction numbers $R_e(\tau)$ for given time periods $\tau$.

We conducted phylodynamic analyses of SARS-CoV-2 sequence data from Denmark, Scotland, Switzerland, and the Australian state Victoria. In analyzing the data we assumed that $R_e^{\text{BEAST}}(\tau)$ was piecewise constant in between major changes in SARS-CoV-2 non-pharmaceutical interventions (intervals stated

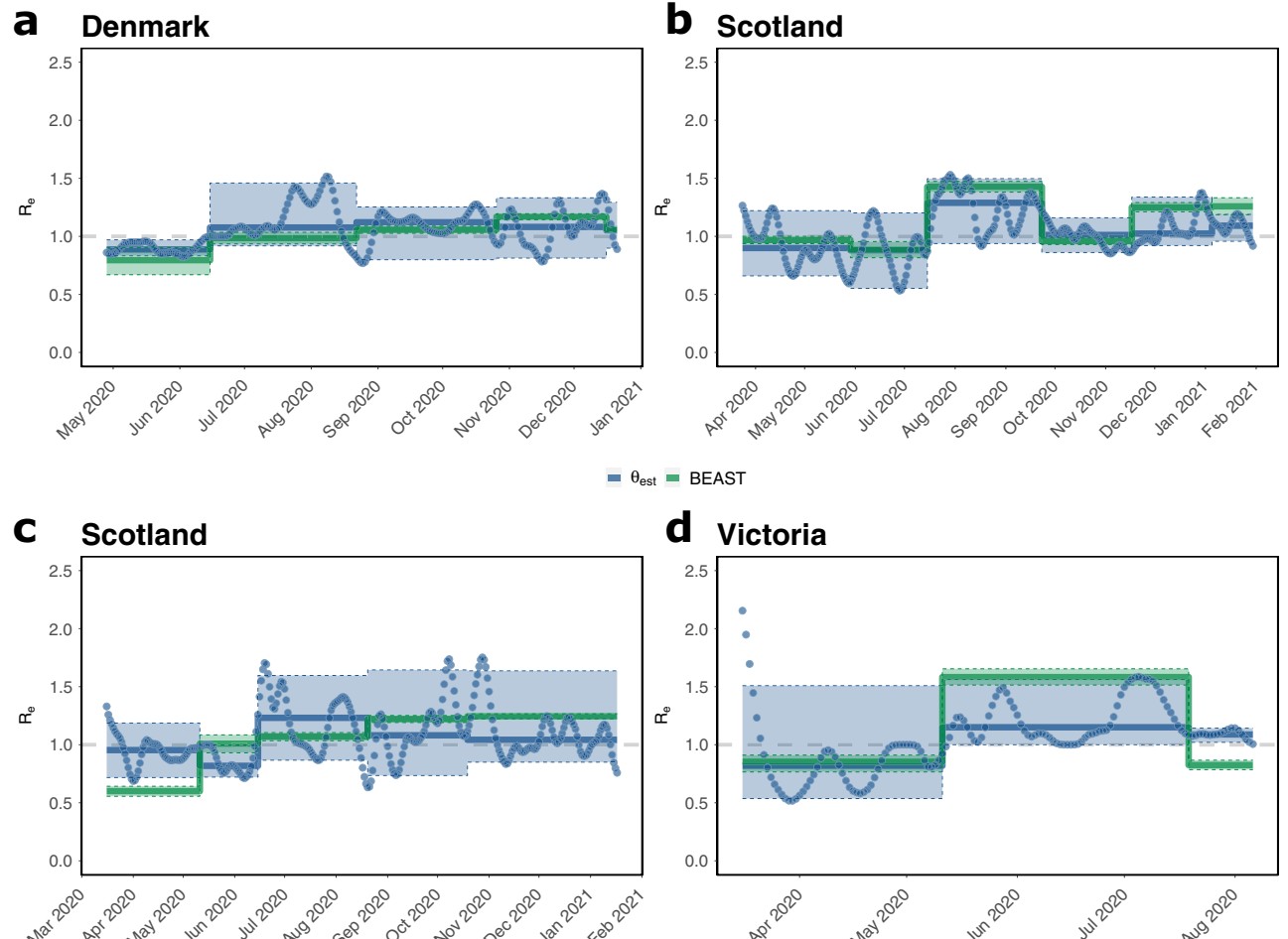

**Fig. 2 Effective reproduction number $R_e$ estimates using the proposed method ($\phi$) and phylodynamics (BEAST2). a–d** Piecewise constant median $R_e^{\text{BEAST}}(\tau)$ estimates (green solid lines) were calculated using the BDSKY model for the indicated intervals, as described in the "Methods" section. Daily estimates $R_e^{\phi}(t)$ (blue dots) were directly calculated from the incidence correlates $\phi$ using the Wallinga-Teunis method[30]. The median of these values for the indicated intervals $R_e^{\phi}(\tau)$ is shown as solid blue lines. The 95% confidence interval is specified by the shaded areas. Justifications of the intervals are found in Supplementary Note 2.

in Supplementary Note 2). We then used BEAST2 to estimate $R_e^{\text{BEAST}}(\tau)$ alongside the tree reconstructions.

In parallel, we estimated corresponding effective reproduction numbers $R_e^{\phi}(t)$ by applying the Wallinga–Teunis method[30] to incidence correlates $\phi$ derived by GInPipe. For both methods, we used publicly available full length SARS-CoV-2 sequencing data from GISAID[23,24] (Supplementary Note 4).

Results of both methods are presented in Fig. 2. Overall, both methods show congruent trends for the analyzed countries, when comparing the piecewise constant $R_e^{\text{BEAST}}(\tau)$ from phylodynamic analysis with the median daily $R_e^{\phi}(t)$ for the same interval. Noteworthy, GInPipe allows for a much finer time-resolution (daily $R_e$ estimates) compared to the piecewise constant $R_e$ estimates on pre-defined intervals, obtained from the phylodynamic analysis.

For Denmark, the first interval spans the decline in the number of infections after the first wave (end of April to mid June). Consequently, we observe $R_e(\tau) < 1$ using both methods. For the next intervals, the median or piece-wise constant $R_e(\tau)$ is predicted to be around, or slightly larger than one. However, GInPipe reconstructs a number of peaks in the daily $R_e^{\phi}(t)$ estimates, most pronounced in August, coinciding with the summer holidays in Europe. In the interval from November to mid December the estimates deviate slightly, with

a larger median estimate from BEAST2, however, both interval estimates are predicted to be $R_e(t) > 1$ and the confidence intervals overlap entirely.

The GInPipe $R_e(\tau)$ estimates for Scotland are within 20% of the corresponding BEAST2 estimates, where GInPipe again allows for a much finer time resolution. Once again, we see a peak in the summer (August–September 2020), coinciding with the summer holidays in Europe. For the last interval (from December 2020) both methods show a median $R_e(t) > 1$, again with a slightly higher median BEAST2 estimate, coinciding with the second wave of infections.

For Switzerland, the estimates disagree slightly, particularly in the first interval (mid March to mid May), which spans both sides of the peak number of infections during the first wave. Although both methods predict a median $R_e(\tau) < 1$, the absolute value differs in magnitude between the two methods, with BEAST2 estimating a much lower value. The lower estimate from the BEAST2-analysis in the first interval may be explained by the approximation of transmission clusters, which results in the reconstruction of a relatively high number of transmission events many of which may have occurred outside Switzerland (Supplementary Note 2, Fig. SN.29 therein, tree B.1). In the daily estimates, we see a transition from $R_e^{\phi}(t) > 1$ to $R_e^{\phi}(t) < 1$, which may explain why the median prediction with GInPipe is close to one for the entire

interval. The estimates are qualitatively different for the second interval (mid May–mid June), where GInPipe estimates $R_e^\phi(\tau) < 1$, while BEAST2 estimates $R_e^{\text{BEAST}}(\tau) \approx 1$. Again, GInPipe estimates a peak in summer (mid June to mid August $R_e\phi(\tau) > 1$). While BEAST2 predicts the onset of transmission in the second wave to already start in mid August ($R_e(\tau) > 1$), GInPipe estimates the first major rise in infections at the end of September.

For Victoria we observe an $R_e^\phi(t) > 1$ until mid March in the daily estimates. Overall, $R_e$ is < 1 for the first interval between mid March and May, versus $R_e > 1$ between June and August. Again, we see various peaks around June and July in the daily $R_e$ estimates with the proposed method. For the final interval, both methods slightly disagree, with $R_e^{\text{BEAST}} < 1$ and $R_e^\phi(\tau) > 1$, though the daily $R_e^\phi(t)$ are decreasing towards the end of the final interval.

In addition to the phylodynamic inference of $R_e^{\text{BEAST}}$, we also implemented phylodynamic incidence reconstruction using EpiInf for Scotland[31]. Incidence trajectories from EpiInf, GInPipe and reported cases are shown in Supplementary Fig. 1. GInPipe estimates the timing of the first (April 2020) and second wave (November 2020) in congruence with the reported cases, while EpiInf estimates the first wave to occur mid May and may underestimate the magnitude of the second wave. In addition, EpiInf estimates a peak in August that is not represented in the reporting data, nor in GInPipe's estimates. With regards to the third wave (January 2021), both EpiInf and GInPipe disagree with the rapid decline seen in the reported cases from January 2021.

In terms of computational time, the entire GInPipe analysis pipeline runs in 25 min on the full Denmark data set ($n = 40.575$ sequences) and in 7 min on the Victoria data set ($n = 10.710$ sequences) on a single notebook (2.3 Ghz, 2 cores). Furthermore, GInPipe does not require to pre-assign any intervals, to exclude particular strains, construct a phylogenetic tree, or cluster sequences based on their phylogenetic relationship. The BEAST2 analysis alone required about 15 h on an Intel Xeon E5-2687W (3.1 Ghz, 2 × 12 cores) on a sub-sampled data set ($n \approx 2500$ sequences) with additional computation time needed to construct a multiple sequence alignment and approximate transmission clusters. Despite recent advances to improve the application of phylogenetic methods to large genomic data sets[32] (https://beast.community/thorney_beast), these methods remain computationally expensive and advanced knowledge is required to apply them properly to bigger data sets.

**Reconstructed incidence histories**. We used GInPipe to reconstruct complete incidence histories for Denmark, Scotland, Switzerland, and Victoria (Australia) from publicly available full-length SARS-CoV-2 sequencing data provided through GISAID[23,24] (Supplementary Note 4). In Fig. 3, we compare the reconstructed incidence histories (blue lines and dots, left axis) to the 7-day rolling average of officially reported new cases (red line, right axis). Overall, the reconstructed incidence estimates reflect the different pandemic waves deduced from the reporting data, although there are quantitative differences between the reconstructed and reported incidence trajectories over time. In particular, during the first wave in Scotland, and Victoria (Fig. 3b, d) our method estimates higher incidences than reported, whereas the curves align at later points for the second and third waves. It is worth mentioning that testing capacities were particularly low in Scotland in April (during the first wave), suggesting extensive under-reporting in the initial phase of the pandemic. This is also supported by test positive rates of almost 40% during April 2020 in Scotland (Supplementary Fig. 2). In Victoria, sufficient testing capacities were not available until May, but test positive rates were already declining from April to May (Supplementary Fig. 2).

This indicates that the first wave may have been under-reported in magnitude, but had vanished by May.

Interestingly, the proposed incidence reconstruction method predicts small summer waves in August in the three European countries (Fig. 3a–c) that are not visible in the reporting data. In the incidence reconstruction method these summer waves are immediately followed by the second SARS-CoV-2 wave. For the second wave, the profiles of the reconstructed incidence histories match the profiles of the reported cases, particularly in Denmark, Scotland, and Victoria (Fig. 3a, b, d). For Scotland, our method predicts a more long-lasting third wave with rising incidence rates until February 2021 and a moderate decline with several smaller peaks until May, whereas the reporting data indicate a peak in January 2021 with a subsequent fast regression. The argument, that ongoing vaccination in Great Britain could explain the immediate decline of reported infected cases, can be objected with the fact, that by March 2021 only about 2% of the Scottish population were fully vaccinated. Moreover, phylodynamic incidence reconstruction using EpiInf[31] (Supplementary Fig. 1) also suggests a more long-lasting third wave in Scotland.

For Switzerland, we predict a larger wave around January–February 2021 (third wave) that is not reflected in the reporting data. Towards the end of the prediction horizon, from March 2021 onwards, the reported cases and the incidence estimation both indicate a rise in numbers (fourth wave).

In addition to the countries analyzed above, we further reconstruct incidence trajectories for Japan, Chile, India, and South Africa for the entire time span from the onset of the pandemic until mid 2021, see Supplementary Fig. 3. They demonstrate GInPipe's ability to reconstruct incidence histories with very limited sequencing data. Particularly for Chile, India, and South Africa, the amount of accessible data are considerably sparser than for the countries analyzed in Fig. 3. All four pandemic waves for Japan and the two major waves for South Africa were reconstructed. For India, and to some extent Chile, the reconstructions indicate sustained high-level spread from early 2020 until February 2021, when the pandemic started to expand massively.

**Relative case detection rate**. We investigated whether the proposed incidence reconstruction method may be used to learn about the proportion of infected cases that are actually tested, detected and reported, $P_t(\text{tested|infected})$.

The proportion of SARS-CoV-2 infected who are actually reported can be calculated using Bayes' formula (see the "Methods" section). In order to perform the calculation, the proportion of actively infected individuals in the population $P_t(\text{infected})$ needs to be known. We have shown that the incidence correlates $\phi$ from our method are proportional to the number of infected individuals, $c \cdot \phi_t = N_{\text{eff}}$ (Figs. 1d–e, 3), and hence to the probability of being infected $P_t(\text{infected})$. Consequently, we may use the reconstructed incidence profiles, together with the test sensitivity and specificity, the respective information about the proportion of positive tests, as well as the testing capacities for each country or region to calculate changes in the case detection rate, scaled by unknown factor $c$.

In Fig. 4, we show the $\log_2$ scaled detection probabilities for Denmark, Scotland, Switzerland, and Victoria (Australia). The log scaling allows us to easily gauge the relative change in (under-) detection of the infected population over time (e.g., twofold, fourfold increase or decrease in case detection rate). The dashed vertical lines in the graphics indicate major changes in testing policies in the respective countries. Individual parameters used in the inference procedure, $P(\text{tested})$, $P(\text{inf|tested})$, and $c \cdot P(\text{infected})$ are shown in Supplementary Fig. 2.

For Denmark, we observe an initial period of massive SARS-CoV-2 under-detection in the beginning of March 2020, Fig. 4a

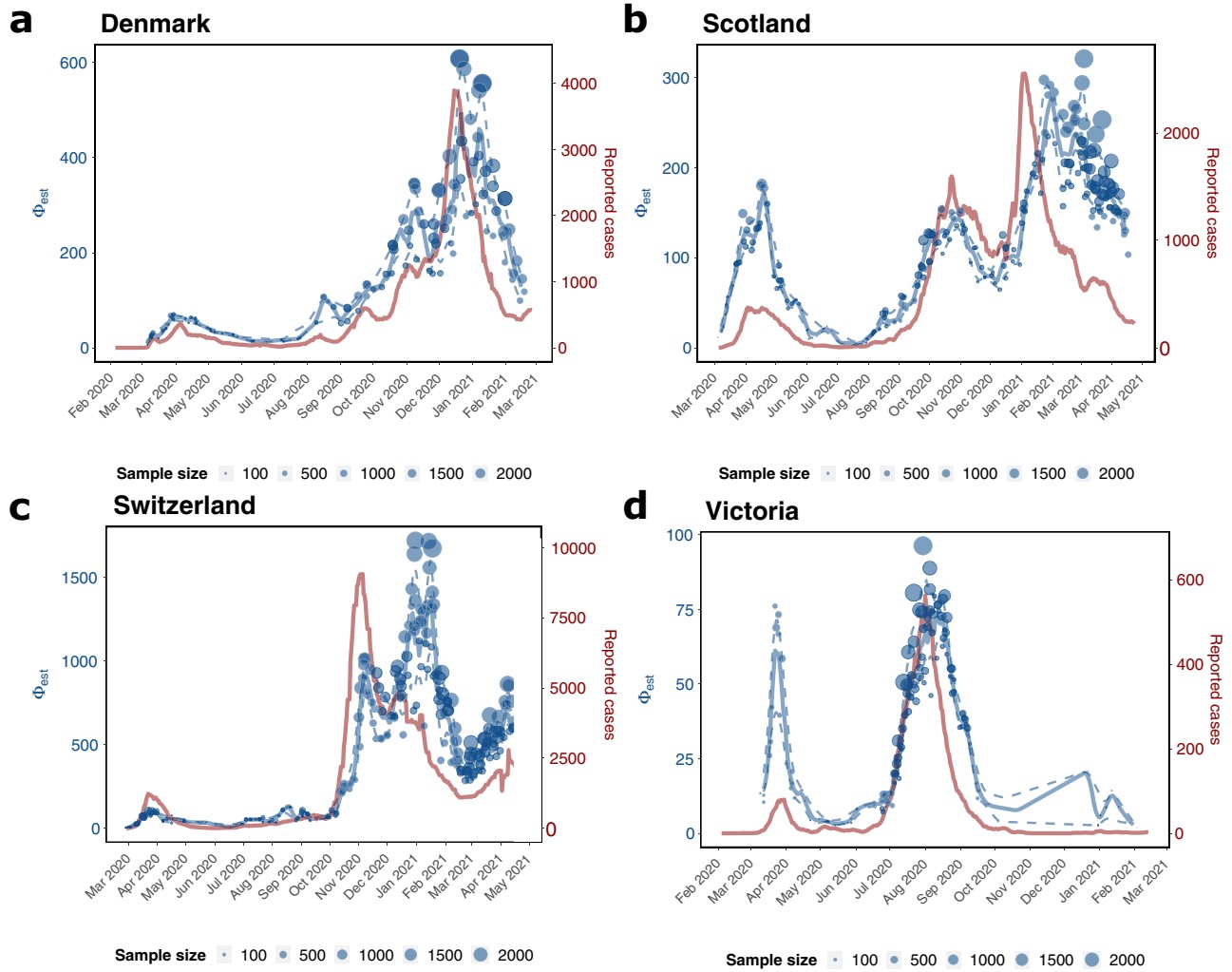

**Fig. 3 Incidence reconstruction based on sequencing data.** The graphic depicts the genome-based incidence reconstruction in blue using the proposed method (left axis) vs. the 7 days rolling average of newly reported cases in red (right axis). Blue dots depict $\phi_b$ point estimates of the incidence correlate, where the size of the dot is related to the number of sequences used to infer $\phi_b$. The solid and dashed blue lines denote the median smoothed trajectories and their 5th and 95th percentiles. The black markers on the x-axis depict the collected sequences at the given dates. **a** Denmark ($n = 40.575$ sequences) **b** Scotland ($n = 30.258$ sequences), **c** Switzerland ($n = 25.779$ sequences), **d** Victoria ($n = 10.710$ sequences).

(upper panel), which coincides with very low testing capacities at the beginning of the pandemic (Fig. 4a, lower panel). From mid March, case detection stabilizes at a sixfold higher level, compared to the first week of March. The second interval begins around mid May with an important policy change, allowing every citizen to get tested without medical referral. Interestingly, compared to the fairly stable case detection levels from mid March to mid May, this policy change leads to a 2–3 fold drop in case detection in the summer months from July to September. Of note, while everybody is granted the possibility to test for SARS-CoV-2, testing capacities remained fairly unchanged (Fig. 4a, lower panel). According to our calculations, the largest proportion of infections remained undetected in July. From end of August, testing capacities were steadily increased in Denmark (Fig. 4a, lower panel), particularly in Copenhagen and at the airports, followed by prioritized testing. From September on, this leads to a nearly eightfold increase of the case detection rate, with a peak in December. From end of December the detection rate drops more than fourold, despite continuous testing.

For Scotland (Fig. 4b), the earliest test data are available only from the end of March. Therefore, the data captures only the second part of the first wave, compare Fig. 3b. In the beginning of

May, testing capacities were more than doubled (Fig. 3b, lower panel) and outbreak investigation intensified. This led to a doubling of the relative case detection rate from May, compared to the first phase. On 18 May, SARS-CoV-2 testing was opened for everyone with symptoms. However, only in July testing capacities were increased. This may have led to a drop in case detection from mid May to July, after which case detection increased and remained during August at roughly the levels achieved in May. After 25 August, testing capacities and accessibility of testing steadily increased. Accordingly, case detection increased about sixfold until winter 20/21. From 25 November, testing capacities were further expanded, especially in the health sector, including hospital patients, health and social care staff, with fairly stable case detection rates. Further increase of testing capacities in the end of December allowed to double the probability to detect infected individuals. From the beginning of the year 2021, the Scottish government pushed community testing in areas with high SARS-CoV-2 prevalence. At the same time, the proportion of positive tests start to decline (Supplementary Fig. 2), and consequently the case detection rate collapses until April by ninefold.

Similar to Denmark, Switzerland shows an initial period of massive SARS-CoV-2 under-detection in the beginning of March

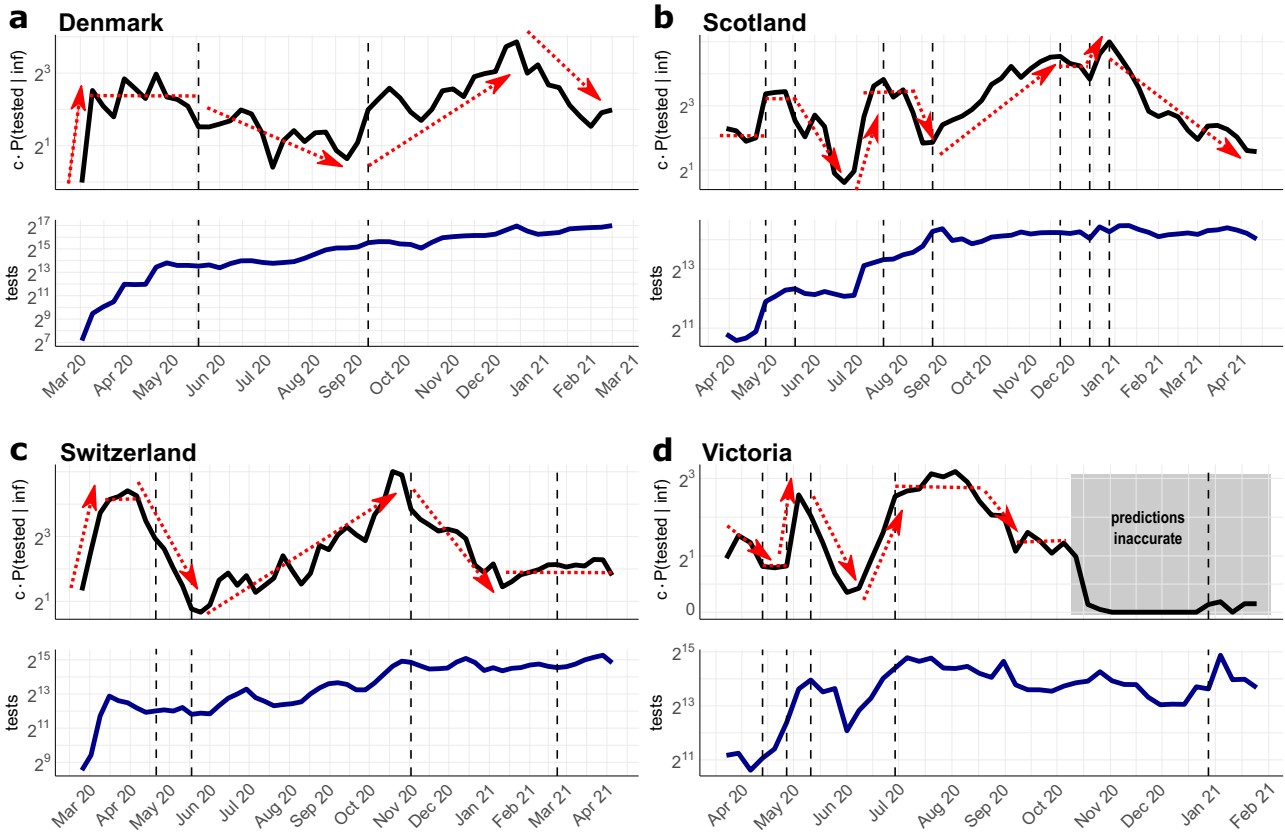

**Fig. 4 Relative case detection rate.** Black line in upper graphics: Estimated and scaled probability of detecting SARS-CoV-2 infected individuals $c \cdot P$(tested | inf). Blue line in the lower graphics: Number of conducted tests per calendar week. Dashed vertical lines indicate major changes in the testing strategies in the respective location. The sources for testing data and strategies are given in Supplementary Note 3. **a** Denmark. Policy changes: 18 May 20: testing for everyone; 9 September 20: increasing testing available, **b** Scotland. Policy changes: 1 May 20: expanded testing strategy including enhanced outbreak investigation; 18 May 20: testing for everyone with symptoms; 22 July 20: including young children for testing; 25 August 20: increasing capacity and accessibility of testing; 25 November 20: expansion of testing in health care; 15 December 20: increase of testing capacity; 1 January 21: community testing in areas with high coronavirus prevalence. **c** Switzerland. Policy changes: 22 April 20: testing for all persons with symptoms; 18 May 20: priority testing; 2 November 20: rapid antigen tests are included in the testing strategy; 27 February 21: recommended preventative and repeated testing as part of precautionary measures. **d** Victoria (Australia). Policy changes: 14 April 20: anyone having symptoms can be tested; 30 April 20: start of 2 weeks testing blitz; 11 May 20: increased surveillance with testing of sewerage; 1 July 20: expanded testing blitzes in outbreak regions; 30 December 21: urging to be tested after re-emergence of positive cases.

2020 (Fig. 4c, upper panel), which coincides with very low testing capacities at the beginning of the pandemic (Fig. 4c, lower panel). When testing capacities increase by mid March, case detection rates grow 8-fold. However, from the beginning of April, we observe a drop in the probability to detect infections that lasts until mid May (overall 10-fold drop). This trend coincides with a drop of positivity rates (Supplementary Fig. 2), as well as the extension of testing criteria on 22nd April: From this date, anybody with symptoms were allowed to get tested, despite the fact that the availability of tests was not increased (Fig. 4c, lower panel). From 18 May, tests were partly prioritized for hospitalized and vulnerable individuals. At the same time, testing capacities steadily increased and incidences dropped. As a net effect, the probability of detecting infected people increases steadily to a maximum at the end of October with a relative difference of nearly 20-fold compared to the low point in mid May. On 2 November, Switzerland begins to supply antigen-based RDT for self-testing as part of their COVID containment strategy. Interestingly, our model predicts that this led to a sharp decline in case detection, again corresponding with the decline in positivity rates (Supplementary Fig. 2). From 21 February 2021, further precautionary actions were taken, and the government recommended repeated testing. This is associated with a stable,

but relatively low detection rate for infected people until the end of April 2021.

For the Australian state Victoria, the earliest data were available from end of March 2020 (Fig. 4d), capturing the second part of the first SARS-CoV-2 wave. Detection probabilities in the first interval, until 14th April were changed proportionally to the test capacities during that interval (Fig. 4d, upper and lower panel). On 14 April 2020, the testing criteria were expanded, allowing anyone with COVID-like symptoms to be tested. Unlike the situation in Switzerland, where we observed a downward trend in case detection after expanding the testing criteria (Fig. 4c), the detection probability in Victoria remains stable until the end of April. In contrast to Switzerland, testing capacities were increased when testing criteria were expanded. On 30 April, the government initiated a two-week testing blitz, a large, coordinated testing campaign to locate viral spread. The testing blitz was accompanied by mass sewerage testing and matched with a massive increase of testing capacities, which led, according to our simulations, to a fourfold increase in the probability to detect infected individuals. At the end of the testing blitz, testing capacities steadily decreased and the proportion of detected infections decreased drastically (by roughly ninefold). At the beginning of June, testing capacities rose again, matched by a rise in the proportion of detected cases. From

1 July onwards, several testing blitzes were conducted in outbreak regions, which seemed to have stabilized case detection rates during the second wave of infections. After the second wave (end of August– September, Fig. 3d), case detection rates drop. From October 2020 onwards, our predictions become highly unreliable, as the incidence estimates credibility interval includes zero (compare Fig. 3d), which concludes that the case detection rate cannot be determined anymore.

In general, we make two striking observations: Firstly, and quite intuitively, whenever more tests were conducted, the proportion of detected SARS-CoV-2 cases increases. Secondly, and unexpectedly, whenever testing criteria were relaxed, this led to a drop in the probability of case detection. We see this drop in mid May in Denmark and Scotland and in mid April in Switzerland. Importantly, the expansions of testing criteria were not-, or insufficiently matched by increased testing capacities. Quite surprisingly, our simulations for Switzerland suggested a drop in case detection when antigen-based RDT self-testing became part of the national diagnostic strategies.

## Discussion

SARS-CoV-2 continues to spread around the world, making epidemiological and molecular surveillance indispensable for the evaluation and guidance of public health interventions.

National and international sequencing efforts are underway that closely monitor the dynamics and evolution of the virus. In the global fight against SARS-CoV-2, many reconstructed sequences have been made broadly available through public databases, such as GISAID[23,24] and the COVID data portal. In this work, we introduce GInPipe, a pipeline that utilizes this data to reconstruct SARS-CoV-2 incidence histories.

Viral infections are often characterized by a transmission bottleneck[33], where only a very small number of viruses initiate the infection and subsequently replicate within the host. A sufficient number of viruses (viral load) is required for further transmission. Hence, the temporal window of infectiousness begins with the intra-host viral population reaching a sufficiently large abundance and ends with the virus becoming eliminated by the immune system (or drugs). In SARS-CoV-2, this window only spans a few days and consequently the virus is almost always transmitted within days after infection, in contrast to HIV, HBV or HCV[25,26]. If neutral or favourable mutations occur during this time, they may become abundant enough to be passed on to other hosts[33]. The consequence is a well-defined duration of intra-patient evolutionary time in which the virus can randomly mutate and become transmitted subsequently. In SARS-CoV-2, this intra-patient evolutionary time appears to be short and the analysis of outbreak clusters indicates that the virus genomes from linked cases were separated by either none, or very few mutations[34–36]. The brevity of evolutionary time before transmission may thus result in relatively homogeneous evolutionary changes between consecutive cases, which would imply strong correlations between evolutionary changes and the number of infections. This evolutionary signal allows GInPipe to reconstruct SARS-CoV-2 incidence histories solely from time-stamped viral genomes.

This presumption may also hold for other respiratory viruses, depending on the rate at which they evolve. In Supplementary Note 1, section SN.1.15, we analyzed whether GInPipe is sensitive to changes in the evolutionary rate. We found that, as long as the evolutionary rate is sufficiently high to produce a measurable evolutionary signal, GInPipe can reliably reconstruct incidence histories.

However, we would expect the method to work less well for sexually transmitted or blood-borne diseases caused by HIV, HBV, or HCV, where the virus can continuously evolve for weeks or years before being transmitted, causing a very heterogeneous signal that may fail to link viral evolution to the number of infections. For example, for a chronic infection like HIV, the consensus sequence in an individual changes over time, even without any onward infections taking place. Therefore, particularly for chronic viral infections, population-level viral evolution is likely both affected by the number of infected individuals, and the generation time of the infection (the average time to pass on the infection)[37].

In the past, numerous approaches have been published, with the aim to estimate the effective population size from genetic properties (reviewed in refs. [38,39]). A variety of methods utilize the information of temporal changes in allele frequency (reviewed in ref. [38]), while others build on population genetic theory and phylodynamic reconstruction[40–42]. GInPipe is inspired by the recent works of Khatri and Burt[27], which has foundations in population genetic theory. Khatri and Burt derived a method to infer the current effective population size with soft selected sweeps from fixated mutations of different origins. They derived a simple function of the mean number of origins and the current allele frequency.

In contrast to ref. [27], we are interested in the history of the effective population size. Therefore, we seek to assess the effective population size per time instance, using time-stamped sequences that are assigned to bins of temporally adjacent sequences. For each bin, we investigate the current population. We utilize the number of haplotypes as an approximation for the mutational input. Akin to the equation in Khatri and Burt[27], the raw evolutionary signal is put in relation to the number of mutants in the data set, which facilitates the usage of the method with incomplete data. Essentially, GInPipe considers snap-shots of inter-patient evolution to estimate a mutational input parameter $\phi(t)$. The latter is proportional to the effective population size, which correlates with the incidence. From the set of time-dependent incidence correlates $\phi_b$, the entire incidence history $\phi(t)$ can be reconstructed.

We assessed the suitability of GInPipe using in silico simulated outbreaks, in comparison with phylodynamics and by comparing to reported case statistics.

Using simulated data, the method robustly reconstructed incidence histories (Fig. 1). It also performed well with incomplete data, and when the sampling rate changed over time (Supplementary Note 1, section SN.1.7–8). Selection within a realistic range had no influence on GInPipe's ability to reconstruct incidence histories (Supplementary Note 1, section SN.1.14). However, we found that GInPipe may become less accurate when too few viral sequences are available. In situations, where the evolutionary signal is diminished, such as biased sampling of sequences by their similarity, the incidence correlates $\phi$ decreases. This can become a problem, if the sampling strategy changes over time, for example from outbreak investigations to more representative sampling strategies. Likewise, if the pandemic was purely driven by a few super-spreaders, the method may not work, because the viruses transmitted from the super-spreader may be clonal, and hence the evolutionary signal may vanish. In essence, if the sampling (and thus the evolutionary signal) is severely distorted, the method will be affected. However, this limitation with regards to biased sampling applies to all methods available (serology, wastewater analysis, phylodynamics, diagnostics). Interestingly, when foreign sequence variants were introduced in the simulations (Supplementary Note 1, section SN.1.12) and subsequently transmitted onward, the performance of the method was not affected. In the extreme case, when we modelled large frequencies (>10%) of introductions, which were not transmitted onward in the community, GInPipe slightly overestimated incidences.

Based on these analyses, we currently view the proposed method as empirical confirmation that for SARS-CoV-2 an evolutionary signal exists, from which the incidence trajectory can be deduced. In the future, we intend to thoroughly advance the theoretical foundation for a precise quantification of a sufficient evolutionary signal. We also plan to make the pipeline more robust by developing filters that recognize abrupt, sampling related, changes in the evolutionary signal that could affect the accuracy of GInPipe. The latter could be achieved by investigating the trade offs between inference of a signal and the temporal resolution of the trajectory.

We also compared the method with epidemiological estimates from phylodynamic reconstruction using BDSKY[28] in BEAST2[29], shown in Fig. 2. Bayesian phylodynamic methods use Monte Carlo Markov Chain (MCMC) or similar techniques to allow for a Bayesian estimation of phylogenetic relatedness of genomes, by both estimating evolutionary parameters, as well as parameters governing an underlying epidemiological model[43,44]. The MCMC sampling procedure makes phylodynamic inference computationally demanding and often requires to down-sample data sets.

When the epidemiological model entails time-varying parameters, changes in the effective reproduction numbers $R_e(\tau)$ can be computed. However, to enable their estimation (practical parameter identifiability), parameters of the underlying epidemiological model are typically considered to be piecewise constant or to change smoothly. In Fig. 2, we show the phylodynamic estimates of the effective reproduction numbers $R_e^{\text{BEAST}}(\tau)$. Corresponding reproductive numbers $R_e^{\phi}(\tau)$ were computed with GInPipe by applying the method of Wallinga-Teunis[30] to the estimated incidence correlates $\phi(t)$. We compared the medians over the temporal windows used in the phylodynamic analysis. Overall, this methodological comparison yielded highly congruent predictions, with the exception of Switzerland in the first- (mid March–May 2020) and final intervals (mid September 2020–January 2021). The ETH Zurich provides a visualization for the daily $R_e$ estimates, based on reporting data (https://ibz-shiny.ethz.ch/covid-19-re-international/). The ETH data, similarly to our daily $R_e^{\phi}$ estimates with GInPipe, shows a peak, followed by a decline in the daily $R_e$ for the first interval. This could explain why the median $R_e^{\phi}$ is only slightly smaller than 1 in this first interval, unlike the BEAST2 estimate, which is ≈0.6. For the final intervals (mid September 2020–January 2021) $R_e^{\phi}$ estimates fluctuate around- or slightly above $R_e(t) = 1$, in line with the predictions of the ETH, and slightly below the BEAST2 estimate that resulted in a median $R_e$ around 1.2. For the sake of this comparison, a relatively crude transmission cluster detection method was employed for the phylodynamic analyses, which may have caused a slight bias in the estimated effective reproduction numbers.

Overall, it appears that both methods yield similar results with respect to inferring the pandemic trajectories in the majority of cases. The power of GInPipe lies in the swift reconstruction of incidence histories with a fine temporal resolution, without requiring phylodynamic inference, construction of a multiple sequence alignment, down-sampling, clustering by, e.g., lineages, or masking of problematic sites in the virus genomes. Moreover, GInPipe performs robustly, even in case of large proportions of introduced variants, which would also include lab-specific errors (Supplementary Note 1). However, $R_e$ estimation is obviously only a side-product of phylodynamic inference, which has many more applications such as the identification and analysis of transmission clusters[45,46], which GInPipe is not suited for. Hence, the two approaches could complement one another.

To simplify the use of GInPipe, we provide an automatic workflow that can be directly applied to data downloads from GISAID or the COVID Data Portal. The execution time appears to scale linearly with the number of sequences to be analyzed (≈1500 sequences per minute on a 2.3 Ghz computer with 2 cores), Supplementary Fig. 4.

When we applied GInPipe to available GISAID data from Denmark, Scotland, Switzerland, Victoria/Australia (Fig. 3), as well as Japan, Chile, India and South Africa (Supplementary Fig. 3), the reconstructed incidence histories appear reasonable when compared with the daily numbers of new reported infections. A remarkable exception is India. For India, the reconstructions indicate a rapid increase in November and high-level ongoing spread from December 2021 until the reported onset of the second wave in early 2021. This time span could coincide with the time point at which the delta variant may have emerged, which is believed to have caused the second massive wave in SARS-CoV-2 infections in India. For Chile, GInPipe estimates fairly stable (high) incidences between July 2020 and February 2021. However, with regards to incidence reconstructions in Chile, sequence sampling between June 2020 and April 2021 is extremely scarce. Accordingly, the uncertainty in GInPipe's reconstructions is large (as indicated by the dashed lines in Supplementary Fig. 3).

For Denmark, reconstructed incidence histories match the reporting data quite well. Of the analyzed countries, Denmark conducted the largest number of SARS-CoV-2 tests per capita (see also $P$(tested) in Supplementary Fig. 2). This could imply that the pandemic was relatively well tracked, as also suggested by relatively small changes in the diagnostic rate (Fig. 4). Moreover, a large fraction of the diagnosed cases were sequenced, providing a comprehensive genomic profile of the virus population.

For the first wave in Scotland and Victoria, we determined a much higher incidence than reported. Notably, the number of SARS-CoV-2 tests per capita was very low in Scotland, as well as in Victoria until May 2020 ($P$(tested) in Supplementary Fig. 2). Thus, a large proportion of infected individuals may not have been diagnosed during this time. In Victoria and Scotland, testing capacities were increased in May, i.e., after the peak of the first wave.

Another striking difference of our predictions in comparison to the reported cases is that GInPipe indicates a rise of infections in August 2020 in all European countries. Notably, this increase in infections coincides with the introduction and community spread of B.1.177 (20E EU1) in most Western European countries as suggested by phylodynamic analyses[47,48]. Our results, when compared with the reported cases, therefore imply an under-reporting of cases during the onset of community transmission of B.1.177.

Quantifying case detection is usually not feasible without knowing, or approximating the proportion of infected individuals (compare Eq. (2)). In order to do so, others have used mathematical models to predict the proportion of infected individuals[5,49] and with this, to estimate the level of under-reporting of SARS-CoV-2. However, these mathematical models cannot be fitted to the reported cases under the presumption of an unknown trajectory of under-reporting. It, therefore, remains extremely difficult to parameterize suitable models for the task of assessing under-reporting, in particular for non-monotonic pandemic trajectories.

Random testing may inform the number of incident, as well as asymptomatic infections[50]. Yet, usually only snap-shots of the incidence may be derived, which are insufficient to parameterize the aforementioned models. Moreover, it is not clear, whether the samples in the random testing scheme were representative. Sero-prevalence studies remain the gold-standard to estimate the cumulative number of infections[5,49], as well as cumulative under-detection. Nevertheless, these studies only provide very coarse time resolution (if any) and require large sample sizes for robust analysis.

A methodologically related approach uses a semi-Bayesian approach to assess under-detection in the US[51]. To enable estimation, the probability of case detection is constrained by the assumption of particular prior distributions.

With regards to the aforementioned approaches, our method to quantify case detecting profiles has the advantage that no complex mathematical modelling is needed, and no constraints are necessary. Instead, we use information about the conducted tests and the test positive rate, in combination with the incidence correlate $\phi$. This makes the proposed approach simple, interpretable and independent of additional assumptions.

Using this method, we observed that broad testing with little, or no suspicion of SARS-CoV-2 infection coincides with apparent under-reporting of infections from the second quarter of 2020. This coincides with a drastic decrease in the proportion of positive test results. From the latter, it is possible to compute the conditional probability that a tested person is actually infected ($P(\text{inf} \mid \text{tested})$, Supplementary Fig. 2). A drop in $P(\text{inf} \mid \text{tested})$ coinciding with a steady amount of tests can negatively affect the probability to detect infected individuals $P(\text{tested} \mid \text{inf})$, which may have happened in the European summer of 2020. In other words, the scarce testing resources available during that time, may not have been employed in the most effective way. This suggests that it may be advisable to focus on testing symptomatic individuals when testing capacity is low.

Nevertheless, the apparent under-reporting was overcome relatively quickly by either increasing testing capacities (Denmark, Scotland, Victoria) or re-focusing capacities or both (Switzerland), Fig. 4. Interestingly, our method predicts a decline in case detection in Switzerland after the broad introduction of antigen self-testing in November 2020. A potential explanation for this observation is that only a fraction of positive antigen self-tests is confirmed by PCR and hence enters the Swiss reporting system. At the time of writing, the final interpretation of this observation is still unclear and will require further analysis.

In summary, we have developed a method that allows to reconstruct incidence histories solely based on time-stamped genetic sequences of SARS-CoV-2. We implemented the method in a fully automated workflow that can be applied to publicly available data. Moreover, this method can be used to assess the impact of testing strategies on case reporting. Finally, we envision that the method will be particularly useful to estimate the extent of the SARS-CoV-2 pandemic in regions where diagnostic surveillance is insufficient for monitoring, but may still yield a few samples for sequencing. In some of these regions pandemic control may be impossible or cause more harm than benefit[52] and hence these regions may constitute reservoirs for the emergence of novel SARS-CoV-2 variants. Gaining insight in the pandemic dynamics in these regions through alternative methods, such as GInPipe, could yield valuable information that helps to direct global SARS-CoV-2 control efforts.

## Methods

**Data and data pre-processing**. Sequences and meta data for Denmark, Scotland, Switzerland, and Victoria (Australia) were downloaded from the GISAID EpiCoV database. Sequences, where only the year of collection was provided were omitted. If year and month are specified, the 15th day of the month was added to the meta data.

The retained sequences were individually mapped to the reference (NCBI Wuhan Reference Sequence: NC_045512.2) with minimap2 version 2.17 (r941), allowing up to 10% of mismatches[53]. From the mapping files (SAM), we deduced the nucleotide substitutions for each sequence. The current version of the pipeline ignores indels. If ambiguous codes contain the reference base, they are replaced by the reference (e.g. 'R' would be replaced by 'A', if 'A' is the reference base). If an ambiguous code does not contain the reference, one of the bases defining the ambiguous code is randomly chosen. In our analysis, point mutations appearing less than three times in the whole data set were filtered out, as they may occur due to sequencing errors[54]. However, this is a user-defined filter in GInPipe. Changing

this filter has a scaling effect on the incidence correlate (changing the slope of the linear correlation).

**Construction of temporal sequence bins**. SARS-CoV-2 sequences were sorted chronologically by collection date and assigned to temporal bins $b$ in a redundant manner. We subdivided the sequence set into bins of

- equal size (proportions of 2, 5, 7% of all samples)
- spanning an equal amount of days (10, 15, and 20, and one calendar week).

In general, the binning strategy should be chosen such that the bins contain enough mutational information (sequences), while allowing sufficient temporal resolution (such that, e.g., peaks and valleys within the population dynamic can still be captured).

In this application, bins that contain a proportion of sequences should span at least 3 days and maximally 21 days, and bins that span a predefined time period should contain at least 15 sequences. The date assigned to a bin is the mean collection date of the comprised sequences.

The redundant binning (re-sampling) allows to evaluate cases where there is insufficient data along the time line (Fig. 1a), and makes the proposed method statistically more robust to outliers, Supplementary Note 1.

**Incidence correlate $\phi_b$**. The proposed method is inspired by the work of Khatri and Burt[27], who derived a simple relation between the mean number of independent origins of soft selective sweeps in a population sample $\bar{\eta}$, the current number of an allele $m$ and mutational input, i.e., the scaled (haploid) effective population size $\theta = 2 N_{\text{eff}} \mu$, with $\mu$ denoting the mutation rate: $\bar{\eta}(t) = \theta \log\left(1 + \frac{m}{\theta}\right)$.

Unlike Khatri and Burt, who aim at estimating the recent effective population size utilizing the recurrent mutations which have been fixated in the population, we seek to reconstruct the history of incidences of a population over time. We adapted the equation accordingly, also under the presumption, that the de novo occurrence of mutations is driven by random chance events, whose likelihood may increase with the number of infected individuals[55,56]. Seeking to estimate the incidence correlates $\phi = c \cdot N_{\text{eff}}$, with the incidence being equivalent to the effective population size $N_{\text{eff}}$, scaled by a constant factor $c$, we parameterize the equation as follows: For each temporal bin $b$ we estimate incidence correlates $\phi_b$ at time $t_b$. From the sequences comprised in bin $b$, i.e. dated within a certain time frame $\Delta d_b$ (Fig. 1a), we infer the number of haplotypes $h_b$ and the total number of mutant sequences $m_b$ in the bin (Fig. 1b). The mutations are determined with respect to a given reference sequence. In the original equation, we replace the mean number of origins $\bar{\eta}$ with the number of distinct variants (haplotypes) $h_b$. In each temporal bin, however, haplotypes and mutants are accumulated over the time span $\Delta d_b$. To correct for biases that result from this accumulation, especially for large time spans, we normalize the inputs $h_b$ and $m_b$ using a logistic function $w_b = \left(\log\left(\sqrt{\Delta d_b}\right) + 1\right)^{-1}$. The parameter $\phi_b$ is derived by numerically solving

$$\phi_b^* = \arg\min_{\phi_b} \quad h_b \cdot w_b - \phi_b \log\left(1 + \frac{m_b \cdot w_b}{\phi_b}\right). \tag{1}$$

**Reconstructing the incidence history**. Incidence point estimates $\phi_b$ are assigned to the mean collection date $t_b$ of the sequences contained in the bin. We applied a moving average filter with window size 7 days to derive a continuous, smoothed trajectory (Fig. 1c). For uncertainty estimation, we sub-sampled $\phi$ trajectories 1000 times, by randomly leaving out 50% of the point estimates and reconstructed the trajectory by smoothing and linear interpolation between the remaining point estimates.

**Effective reproduction number $R_e$**. Based on the reconstructed incidence histories, the effective reproduction number $R_e(t)$ was computed using the established method by Wallinga and Teunis[30] (R package R0 version 1.2.6[57]). Daily estimates of $\phi$ were assigned a pseudocount of one and rounded to the nearest integer. For the generation time distribution $g(\tau)$ of SARS-CoV-2, we chose the Gamma distribution with a mean of 5 days and a standard deviation of 1 day[58,59].

**Simulation study**. To evaluate the proposed incidence reconstruction method, we generated test data by stochastically simulating the evolutionary dynamics of a viral outbreak using a Poisson process formalism. We started with $N(t_0) = 50$ copies of a random sequence of nucleotide length $L = 1000$, that evolved in 120 discrete time steps, depending on a population dynamic. A succeeding generation was modelled to consist of $N(t + 1) \sim \text{Poiss}(N(t) \cdot \rho(t))$ sequences (=effective population size), where we chose a sinusoidal rate $\rho(t) = \frac{\sin(t \cdot 0.11)}{15} + 1.02$. Thus, $N(t + 1)$ sequences from the actual generation were randomly chosen with replacement and copied over to the next generation. We then introduced $n_{\text{mut}} \sim \text{Poiss}(\mu \cdot N(t + 1) \cdot L)$ random mutations into these sequences with per site mutation rate $\mu = 0.0001$.

For each generation, a fasta file with all sequences was stored and used as input for the incidence reconstruction pipeline. We ran 10 stochastic simulations with

the settings stated above to compared the ground truth effective population sizes $N(t)$ from our simulations with the corresponding inferred incidence trajectories $\phi$.

In Supplementary Note 1, we evaluated scenarios where only a fraction of the sequences were sampled (10–90%) and, to rule out sampling biases, we sub-sampled equal amounts of sequences at each time point, independent of $N(t)$. In addition, we assessed the effects of similarity-biased sampling, as well as time-dependent sub-sampling. In further evaluations, we examined GInPipe's performance with simulations in continuous time, non-smooth dynamics with sudden changes of population size, under selective pressure, and with a range of mutation rates. Moreover, we assessed whether our predictions were affected by the introduction of unrelated sequence variants into the population.

**Phylodynamic analyses.** Phylodynamic analyses were performed on subsampled sets of the data described above using a birth-death-sampling process as implemented in the BDSKY[28] model (version 1.4.6) in BEAST2[29] (version 2.6.3) with BEAGLE 3.1.2. Here the precise collection day of sequence samples with only information on year and month was inferred during the analysis and not a priori set to the 15th. The full data sets were first grouped by Pango lineage (Pangolin version 2.3.8)[60,61] and then subsampled by randomly selecting a specific percentage of sequences per week to speed up the analyses (Victoria: 10% for lineage D.2, 50% for other lineages; Switzerland: 50% for all lineages; Scotland: 20% for all lineages; Denmark: 5% for all lineages). In Victoria, lineage D.2 constitutes more than 80% of all sequences in the original data set. Hence, we used a smaller percentage of D.2 lineages, to retain sufficient non-D.2 lineages after subsampling. In addition, sequences were excluded if they belonged to a lineage with less than two representatives in the analyzed set and to ensure separation of consecutive epidemic waves lineages with periods longer than 75 days without any sample were split into separate clusters. Retained sequences were aligned to the reference genome (Genbank-ID MN908947.3[62]) in MAFFT[63] (version 7.453) using the *–keeplength* option and problematic sites were masked by replacing them with N in the alignment[64].

For each remaining approximate cluster a separate phylogeny was reconstructed. A strict clock model with a fixed rate of $8 \times 10^{-4}$ substitutions per site per year and an HKY substitution model were used. In the embedded transmission model, transmission ($\lambda$), recovery ($\mu$) and sampling ($\psi$) rates were assumed to be piecewise constant with changes allowed either when intervention measures changed, or in a uniform manner (Supplementary Note 2). The reproductive number $R_e(t) = \lambda(t)/(\mu(t) + \psi(t))$ was drawn from a log-normal distribution $R_e(t) \sim \log \mathcal{N}(0, 4)$, the rate to become non-infectious $\delta(t) = \mu(t) + \psi(t)$ from a narrow normal distribution with $\delta(t) \sim \mathcal{N}(27.11, 1)$ which is changed to $\mathcal{N}(48.8, 1)$ after first control measures are implemented in the respective area. The sampling proportion $s(t) = \psi(t)/(\psi(t) + \mu(t))$ was a priori assumed to arise from a uniform distribution with a lower limit of zero and the upper limit determined by the ratio of analyzed sequences over diagnosed cases $s \sim U(0, q_i/d_i)$ where $d_i$ is the number of diagnoses and $q_i$ the number of sequences included in the analysis in interval $i$. To account for the lineage-specific subsampling, a separate sampling proportion for lineage D.2, $s_{D.2}$, was modelled in the analysis of the Victoria data. A uniform distribution with an upper limit corresponding to the subsampling percentage was thus used as prior distribution of the D.2 specific-, as well as general sampling proportion $s_g$, i.e., $s_{D.2} \sim U(0, 0.1)$ and $s_g \sim U(0, 0.5)$. Setup files for all four analyses can be found as Supplementary Files.

MCMC chains were run until all parameters converged, which took about 300 million steps for analyses of data from Denmark, Scotland and Switzerland. Because of the large D.2 cluster consisting of more than 900 sequences, about 750 million steps were needed for convergence using data from Victoria. On an Intel Xeon CPU E5-2687W (3.1 Ghz; $2 \times 12$ cores), this corresponded to about 15 h to run one analysis for at least 300 million MCMC steps (about 3min/Msample). Log files were assessed using Tracer[65] (version 1.7.1) and are included as Supplementary Files. TreeAnnotator (version 2.6.0) was used to summarize the posterior sample of phylogenetic trees to a maximum clade credibility tree using median node heights. Lineage through time plots of all summary trees were calculated using the R package ape[66] (version 5.4-1) and are shown in Supplementary Note 2.

To reconstruct daily incidences for Scotland as shown in Supplementary Fig. 1, epidemic trajectories over time for each approximate cluster were simulated using the particle filtering approach EpiInf[31] (version 7.5.2) using the rates inferred during the BDSKY runs with 1000 particles. The full area-specific incidence was then calculated as the sum of all cluster-specific incidences and scaled to represent daily incidence estimates.

**Relative case detection rate.** We used GInPipe to infer changes in SARS-CoV-2 case detection. Let us denote by $P_t(\text{tested}|\text{infected})$ the proportion of infected individuals that are actually diagnosed with the virus in week $t$. According to Bayes' theorem we have

$$P_t(\text{tested}|\text{infected}) = \frac{P_t(\text{infected}|\text{tested}) \cdot P_t(\text{tested})}{P_t(\text{infected})} \quad (2)$$

where $P_t(\text{infected}|\text{tested})$ denotes the proportion of tested individuals that are infected, $P_t(\text{tested})$ the proportion of individuals that are tested and $P_t(\text{infected})$

the proportion currently infected in week $t$. We calculate $P_t(\text{infected}|\text{tested}) = \frac{r_{\text{pos}} - (1 - \text{spec})}{\text{sens} - (1 - \text{spec})}$ from the positivity rate $r_{\text{pos}}$ of the conducted tests, corrected for the clinical sensitivity $\text{sens} = 0.7$ and specificity $\text{spec} = 0.999$ of the diagnostic tests[67]. For calculating the probability of being tested $P(\text{tested})$, we considered linear-, Poisson-, and Binomial models, all of which yielded identical results. For all illustrations herein, we used the latter, yielding $P_t(\text{tested}) = 1 - (1 - 1/\text{pop})^{n_t}$, with pop denoting the population size in the respective regions or country and $n_t$ denoting the number of tests conducted in the respective week.

The probability of currently being infected $P(\text{infected}) \approx \frac{N_{\text{eff}}}{\text{pop}}$ is unknown. However, since we know that $N_{\text{eff}}$ is linearly correlated with the incidence estimate $\phi$, we have $P(\text{infected}) \approx c \cdot \frac{\phi}{\text{pop}}$. Putting everything together we can estimate the relative case detection rate:

$$P_t(\text{tested}|\text{infected}) \cdot c = \frac{\text{pop}}{\phi_t} \cdot \frac{r_{\text{pos}} - (1 - \text{spec})}{\text{sens} - (1 - \text{spec})} \cdot \left(1 - \left(1 - \frac{1}{\text{pop}}\right)^{n_t}\right).$$

Sources for the weekly number of performed tests, as well as test positive rates are stated in Supplementary Note 3.

**Reporting summary.** Further information on research design is available in the Nature Research Reporting Summary linked to this article.

## Data availability

SARS-CoV-2 sequences were downloaded from the GISAID EpiCoV database[23,24] (www.gisaid.org; accession codes in Supplementary Note 4). The Wuhan Reference Sequence was downloaded from NCBI, accession number NC_045512.2[68]. Source data are provided with this paper.

## Code availability

All methods were implemented in Python version 3.9 and R version 4.0. A fully automated workflow has been generated using Snakemake version 6.6.1[69] and is available from https://github.com/KleistLab/GInPipe[70].

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

## Acknowledgements

The authors acknowledge all labs contributing SARS-CoV-2 sequences to the GISAID EpiCoV database as stated in Supplementary Note 4. We thank Timothy G. Vaughan for helpful discussions about EpiInf. M.R.S., M.T., Y.D. and MvK acknowledge funding from the Germany ministry for science and education (BMBF; grant numbers 01KI2016 and 031L0176A). D.K. and A.W. acknowledge funding from the Max Planck Society. A.W. acknowledges financial support through a scholarship (Landesgraduiertenstipendium), funded by the State of Thuringia, Germany. The funders had no role in designing the research or the decision to publish.

## Author contributions

Conceptualization, M.R.S., M.T. and m.v.K.; Methodology, M.R.S., M.T., A.W., D.K. and M.v.K.; Investigation, M.R.S., M.T., A.W. and Y.D. Writing - Original Draft, M.R.S., M.T., A.W. and M.v.K.; Writing-Review and Editing, M.R.S., M.T., A.W., D.K. and M.v.K.; Funding Acquisition, A.W., D.K. and M.v.K.; Supervision, D.K. and M.v.K.;

## Funding

## Competing interests

The authors declare no competing interests.
