## [Peer Review File · Nature Communications]

Reviewers' Comments:

Reviewer #1:

Remarks to the Author:

This manuscript presents a new method for estimating the number of SARS-CoV-2 infections over time from dated genome sequence data. There is an enormous number of genome sequences that have been collected from multiple sites around the world, with nearly 2 million currently available through GISAID. Making effective use of these data in a timely manner is critically important. However, the state-of-the-art — arguably Bayesian phylodynamics — does not readily scale to such numbers (notwithstanding the latest improvements to such methods that specifically address this limitation). Thus, the method described in this manuscript is a welcome addition to our analytical toolkit for the current pandemic and for the future.

The method itself is quite simple. However, the authors' explanation of the method is not adequately clear. It is based on a result from theoretical population genetics, specifically a recent analysis of soft selective sweeps by Bhavin Khatri and Austin Burt. First, the authors are making an analogy between SARS-CoV-2 incidence and a selective sweep due to the simultaneous, deterministic growth under positive selection of multiple lineages of independent origins (and that are also fated to reach fixation with probability $2s$), carrying the same mutant allele on different genetic backgrounds. This analogy is not made explicit and requires a careful explanation. The exact interpretation of "number of mutant sequences" (m) and "number of haplotypes" (h) is unclear. Since the authors' application of the model to incidence does not seem to focus on any particular mutation (relative to a reference genome), does m represent the absolute number of sampled infections (irrespective of sequence) in a given time period (indexed by t)? In other words, does m adjust for sampling effort? Does h represent the number of unique genome sequences? If so, how do the authors deal with ambiguous or base calls or incomplete sequence coverage? Figure 1 does help visually explain h and m to some extent, but there needs to be a clearer explanation and rationale integrated into the main text. The term "number of mutant sequences" is particularly confusing.

In addition, their analogy appears to interpret the uninfected susceptible population as wild-type alleles in a population of constant size. The population dynamics of SARS-CoV-2 does not resemble a selective sweep. What are the consequences of non-sigmoidal dynamics in the number of infections? Is the origination of the mutant allele in a new haplotype analogous to the importation of SARS-CoV-2 from an external source to the population, and does this stipulate that the imported infection carries a unique genome sequence? The mutation rate is incorporated into the derivation of Khatri and Burt's result, but it does not make sense if origination corresponds instead to an importation process. I had expected to see these issues addressed substantially in the Discussion, but most of this section was used to review study results rather than discussing the model assumptions and limitations.

The manuscript presents some simulation results, which is a necessary step for validating a new method, since the ground truth is known without error. Population dynamics were simulated by drawing from a Poisson distribution centred on the population size at the previous time point with a deterministic sinusoidal coefficient driving variation over time, instead of a more parametric model (such as an epochal SEIR model). I am somewhat concerned that the authors were not sufficiently critical of their model with respect to its sensitivity to incomplete sampling and importation of cases. For example, incomplete sampling was assessed by censoring infections completely at random (or stratified by time window), but systematic associations between variation in sampling rates and genomic variation (for example, concentrated sampling of a particular district or subpopulation) may induce a more serious bias. Additionally, the impact of importation on model estimates was simulated by adding genomes in which 10% of sites were randomly mutated with respect to the "founder sequence of the local outbreak". This is an excessive amount of mutation. It is not apparent to me whether this simulation setting is meant to induce a large effect, i.e., make a conservative assessment on sensitivity of the method to importation. Lastly, percent deviation from linearity (supplementary figures) is difficult to interpret as a quantitative outcome of the simulation experiments.

Having raised these issues, I am nonetheless quite impressed with the method presented in the

manuscript - it seems to work surprisingly well on my test data. I think this will be an important contribution not only to the field of molecular epidemiology, but also for public health applications of sequence analysis. It might be helpful to quantify how much more we learn about the number of unsampled infections from this sequence analysis in comparison to conventional data sources such as test positivity rates, if possible.

Running the program:

I was able to install and run demo code on both macOS Catalina 10.15.7 and Ubuntu 18.04.5. However, I ran into problems when attempting to run GInPipe on a custom data set comprising about 5,000 SARS-CoV-2 genome sequences. First, the snakemake workflow had problems dealing with a relative path to the reference FASTA file in the configuration YAML (the program threw the following exception: "MissingIndexException: Missing input files for rule minimap_index_ref"). I had to move the YAML file into the nested directory and use the filename without any relative path prefix. This input specification needs to be more flexible. Next, I ran into a ValueError exception with the error message "invalid contig" when running the pipeline with NC_045512 as the reference genome. Replacing the sequence header with ">ref" seems to fix this problem, but there is no such requirement specified in the documentation. Overall, I found the pipeline to be quite unforgiving about path specifications and the locations of input files.

The third exception I encountered was associated with "rule run_binning", with "CalledProcessError in line 145". This seems to be associated with an error in the Python script "sam_to_bins_modular.py" on line 260 with "KeyError: 0". At first, I suspected that this was due to one or more incomplete sample collection dates in the inputs. However, I found no such instance when grep'ing the input files. I then realized that the problem was that I had included spaces around the pipe ('|') delimiter between sample name and collection date fields. My reason for doing so was because the README documentation actually includes a space between "some_name" and the pipe character. Removing the excess spaces resolved this issue. Hence, the documentation needs to be more explicit about how sequence headers in the sample FASTA input should be formatted. Afterwards, I was able to run the pipeline to completion on these data. The locations of peaks in the incidence correlate plot was generally consistent with the first and second waves (with respect to daily numbers of confirmed cases) for the region represented in the data. Since these trends are fairly correlated with sample collection dates (i.e., more samples collected during waves), I also re-ran the analysis with a random permutation of collection dates among sequences to confirm that the same incidence correlate trend could not be recovered. I didn't have time to run more extensive tests.

Source code:

- I appreciate that the authors have released their source code into the public domain under a permissive license (GPLv3). The Python code looks fairly PEP8 compliant.
- Some of the Python scripts are rather unstructured, in that the code is seldom modularized into functions, e.g., `fix_cigars_subprocess.py`. This makes it somewhat more difficult to interpret the code, and prevents users from adapting the functionality of GInPipe into other workflows in a modular fashion. (Same goes for the R scripts - could the developers please consider turning these scripts into a package?)
- Some of the code style is unconventional. For example, the developers make frequent use of string concatenation instead of Python's built-in methods for formatted strings, such as `str.format()` or C-style formatted strings (with '%' placeholders).
- External programs are being run through the shell, which is generally considered bad practice. For example, a user might be exposed to a shell injection attack if they ran a YAML configuration file with malicious text passed to snakemake parameters. Recommended method is `subprocess.check_call()`.

- please consider using temporary files via Python module tempfile rather than writing to hard-coded file names like `list_of_files.tsv`.
- clearing the user's workspace with an `rm()` command in the R script is not really user friendly, particularly if a user sources one of these files in an interactive R session.
- many functions in the R scripts need documentation; code style is a bit inconsistent, e.g., varying use of `=` and `<-` assignment operations, varying use of whitespace.
- bam_to_fingerprints.py, lines 103-127 would be more readable code if you used enumerate to iterate over cigar, and then unpacked the tuple into variables, i.e.,

```
for i, cigtuple in enumerate(cigar):
    operation, length = cigtuple
    if operation == 0: # and so on
```

Specific comments:

- generally, the manuscript is in a very inconvenient format for review (single-spaced, narrow margins, no line numbering)
- when installing GInPipe on macOS Catalina 10.15.7, I also had to install mamba in order for `conda` to detect `snakemake`, whereas I did not encounter this problem in Ubuntu 18.04, so this doesn't seem to be a Linux-specific issue as implied by the README document.
- the R package mgcv is used in `splineRoutines.R` - shouldn't this be listed as a dependency?
- "the sequences are placed into temporal bins b " - this is awkward phrasing, are these bins indexed by variable b , or is b the total number of bins?
- p.2, please clearly define "mutant sequences" and "haplotypes" at first use
- p.2 "point estimates are prone to slight underestimation" Please provide quantitative results instead of a qualitative summary.
- p.3, regarding BEAST2, there are some recent advances that should enable users to run larger, low diversity (e.g., SARS-CoV-2) datasets than before, such as PIQMEE
- Figure 1A, y-axis label - why not just say "cumulative number of sequences" instead of using a formula that may frustrate some readers?
- the variant filtering step did not seem to exclude any sequences for either the demonstration data or my own data.
- page 4, " $R_e(\tau)$ estimates for Scotland agree almost exactly" Please provide quantitative results.
- Figure 3, since incidence estimates ϕ are correlated, the relation between the two scales (ϕ and reported cases) is arbitrary. How did you decide on a proportionality constant for drawing data on these two scales?
- Figure 4, space permitting, it would be helpful to directly label the vertical dashed lines that correspond to different policy changes.
- pages 6-7, much of the text here is essentially describing features of Figure 4; I think this word count would be better invested in describing and discussing the underlying method (i.e., adapting Khatri and Burt's method).

- page 8, "the vast majority of reconstructed sequence data has been made broadly available through public databases" Unfortunately this is only true for a minority of countries such as Denmark and the UK.
- page 9, "The power of GInPipe lies in the swift reconstruction [...] without requiring [...] masking of problematic sites in the virus genomes." This is not a computationally expensive step and benefits from domain expertise, so why not make use of this filtering step in pre-processing?
- page 9, "The execution time appears to scale linearly with the number of sequences to be analyzed" It would be appropriate to provide some actual results here in supplementary material.
- page 10, "Point mutations appearing less than three times in the whole data set were filtered out, as they may occur due to sequencing errors." This is a problematic assumption. Depending on the size of the data set, a large number of biologically real mutations will fall below this frequency threshold. How sensitive are the results to relaxing this threshold?
- page 10, "we deduced the nucleotide substitutions for each sequence" - so this method excludes indel polymorphisms? Is this justifiable?
- page 11, what convolution filter, exactly?

signed,
Art Poon

Reviewer #2:

Remarks to the Author:

The authors propose a novel method (GInPipe) to estimate the true incidence of SARS-CoV-2 using time-stamped viral genomic data. By analyzing the number and frequency of sequence variants at a given time, they are able to estimate the effective reproductive number and the relative incidence of infection. They validated this method using in silico data, simulating various scenarios including missing/incomplete genomic data, and the introduction of new variants into the population. Subsequently, they validated their model against real-world data from 4 countries: Denmark, Scotland, Switzerland and the Australian state of Victoria. They compared the estimates for R_e from BEAST versus GInPipe as well as relative incidence versus the actual number of reported cases in each country/region.

Overall, the manuscript is well written and represents a comprehensive validation of a complementary method to estimate COVID-19 disease incidence. This method will be especially useful when more sensitive diagnostic tests are inadequate relative to the extent of the outbreak. However, it does require the availability of a significant amount of genomic data, which is usually only available in countries with sufficient resources for PCR and sequencing. That said, there are important observations that can be inferred from their analysis - when the availability of PCR testing is reduced because of a perceived reduction in the number of cases, the genomic data from those cases may reveal more widespread, cryptic transmission; and while there is utility of rapid antigen testing, widespread use of this less sensitive method may underestimate the true incidence of disease as indicated by genomic data.

It is not clear why the 4 datasets (Denmark, Scotland, Switzerland, and Victoria) were chosen. The a priori rationale for choosing these datasets needs to be stated and justified. This is important for the real-world validity of their results.

The mutation rate is not constant throughout the SARS-CoV-2 genome. There are regions under neutral pressure whereas other regions are under selective pressure. In addition, there are synonymous and non-synonymous mutations. Could the method be improved by using only neutral regions of the genome and/or non-synonymous mutations? Could the authors explain the rationale for grouping by Pango lineage and subsampling within lineages?

Reviewer #3:

Remarks to the Author:

The aims of this paper – to approximate incidence using genetic data alone and to compute changes in the probability of reporting are both important and interesting. Characterising the incidence of cases and even deaths is not simple, especially in the face of detection delays and under-ascertainment. An approach that can circumvent some of these problems would be a valuable addition to the outbreak response toolkit. This paper makes some good progress towards these aims but I have several major concerns around validation and accuracy, which need to be resolved for this analysis/methodology to be convincing.

1. The validation on simulated data is not yet sufficient. This is especially important for a paper proposing a new method. A couple more examples with different dynamics should be included and then some statistics computed to showcase accuracy (e.g., considering the lag and scaling between the true incidence and inferred correlate). In particular, the current example shows clear differences ($t = 30-50$ and $t > 100$) that need to be explained and accounted for before the claim of accuracy can be upheld.

2. The approach to simulated epidemics also seems somewhat strange (especially given the use of the Wallinga-Teunis method later). Why not use a renewal model to more accurately simulate what an epidemic might look like (and which is the model behind the Wallinga-Teunis)? The key difference from the current approach would be the use of a generation time distribution (which is better suited for properly considering incidence on daily scales as the paper provides) rather than a simple branching process with fixed generations.

3. The comparisons of R_e via BDSky and the Wallinga-Teunis approach do not seem that consistent – more analysis is needed, and the confidence intervals of both approaches do not seem that clear. While the need for piecewise constant R_e from BDSky is understandable, there still are discrepancies that warrant a closer look.

4. Why not also compare the N_e with coalescent approaches? It does not appear the N_e from the method chosen has been considered against more standard approaches such as <https://academic.oup.com/mbe/article/22/5/1185/1066885>. It would be good to know if the correlation between N_e and incidence is general.

5. The methods of <https://royalsocietypublishing.org/doi/full/10.1098/rstb.2010.0060> have explicitly investigated relationships among N_e and prevalence/incidence. I think this paper should comment on those links since it proposes another correlation.

6. In the supplement the importance of binning strategies is noted. Can some comment in the main text be given for what selection approach was taken? Is there some good theoretical reason? The bias-variance trade-off of bins is well known at least for N_e <https://academic.oup.com/sysbio/article-abstract/68/5/730/5307781>. Can some related comment be made in the choices of this approach?

We hypothesize that the genetic data alone holds information about the pandemic trajectory – I would remove this (as it is what makes phylodynamics as a whole useful) and go to the next line, which is the actual hypothesis specifically examined here.

The approach builds on recent work by Khatri and Burt... – could you add a line with some additional explanation here to improve readability for those unfamiliar with this paper? This is particularly helpful since this is a major point underlying the paper.

We observed a strong ($r = 0.96$)... This correlation is not as informative as it could be. A similar association but done per time point would be more useful to confirm if the seeming lag between the inferred and true N_e is upheld or an artefact. Such lags are important for a method providing incidence estimates given what of the key differences between incidence and reported cases is

indeed the lag, the influence of which has been debated. E.g., see <https://journals.plos.org/ploscompbiol/article?id=10.1371/journal.pcbi.1008409>

Our analyses showed that the method can still accurately reconstruct incidence histories over time, when data is missing or when data sampling is unbalanced – this needs to be better explained and qualified/validated. It seems counterintuitive given that sampling is well known to be a major source of bias both in genetic data and case data (and for estimating either R_e or N_e). If this claimed robustness does hold then it is worth including background for why this would be an advance/important trait of the method e.g., for case data/ R_e see <https://academic.oup.com/aje/article/178/9/1505/89262?login=true> and for genetic data/ N_e <https://academic.oup.com/mbe/article/37/8/2414/5719057?login=true>

Finally, we evaluated whether introductions of foreign sequences affect the reconstruction of incidence histories – this is another counterintuitive point since introductions/imports affect estimates of key epidemiological parameters as has been found across COVID-19. I think this needs more qualification and detail.

For the second wave, reconstructed incidence histories correspond to the reported cases – this does not seem quite right as reported cases themselves do not correspond with the incidence. Please clarify what should be comparable.

Taken together, these lines of evidence suggest that evolutionary change of SARS-CoV-2, the effective viral population size, and the number of infected people are correlated – could some more detail and intuition be provided to help readers understand why this correlation, which is the main assumption behind the method, is valid?

Finally, we envision that the method will be particularly useful to estimate the extent of the SARS-CoV-2 pandemic in regions where diagnostic surveillance is insufficient for monitoring, but may still yield a few samples for sequencing – has this point been demonstrated as possible?

The reproductive number $R_e(t)$... was drawn from a log-normal distribution ... which is changed to $N(48.8;1)$ after first control measures are implemented in the respective area – can some more intuition and explanation be provided for these choices?

We thank the reviewers for their constructive comments, which we believe further improved the manuscript. We particularly appreciate the speed in which they delivered their feedback. Based on the reviewers comments, we performed extensive additional experiments. Below is a point-by-point response to all reviewers' comments and a documentation of all changes and additional experiments. All changes are marked in purple in the 'manuscript with track changes' and page and line numbers in the response letter refer to the 'manuscript with tracked changes'.

Major changes:

- In response to reviewers #1-3, we performed extensive tests of GInPipe, which are documented in the extended Supplementary Note 1.
- In response to Rev. #1, we rewrote parts of the discussion and streamlined the tool.
- In response to reviewers #2-3, we analysed further countries (Japan, Chile, South Africa, India), documented in Supplementary Figure 3.
- In response to reviewer #3, we performed further phylodynamic analysis to estimate incidence. Unfortunately, setting up and performing this analysis consumed most time (as stated in the manuscript, setting up and performing phylodynamic analysis requires considerable expertise and computational time to derive meaningful results). We were however able to derive phylodynamically reconstructed incidence profiles for Scotland (using EpiInf), which are shown together with reported cases and the results of GInPipe in Supplementary Figure 1.

We hope that the reviewers are content with and convinced by the additional analyses.

REVIEWER **COMMENTS**

Reviewer #1 (Remarks to the Author):

This manuscript presents a new method for estimating the number of SARS-CoV-2 infections over time from dated genome sequence data. There is an enormous number of genome sequences that have been collected from multiple sites around the world, with nearly 2 million currently available through GISAID. Making effective use of these data in a timely manner is critically important. However, the state-of-the-art — arguably Bayesian phylodynamics — does not readily scale to such numbers (notwithstanding the latest improvements to such methods that specifically address this limitation). Thus, the method described in this manuscript is a welcome addition to our analytical toolkit for the current pandemic and for the future.

-> Many thanks for the very constructive feedback and also for the time that went into the extensive testing of the pipeline. Apologies if the reviewer felt that the pipeline was so unforgiving in its initial state. We are grateful to make the tool more user- and developer-friendly.

1. The method itself is quite simple. However, the authors' explanation of the method is not adequately clear. It is based on a result from theoretical population genetics, specifically a recent analysis of soft selective sweeps by Bhavin Khatri and Austin Burt. First, the authors are making an analogy between SARS-CoV-2 incidence and a selective sweep due to the simultaneous, deterministic growth under positive selection of multiple lineages of independent origins (and that are also fated to reach fixation with probability 2s), carrying the same mutant allele on different genetic backgrounds. This analogy is not made explicit and requires a careful explanation.

-> Thank you for this critical assessment. We have not clearly pointed out that while the method is **motivated by** Khatri & Burt's article, it is actually quite different regarding the points made by the reviewer. We have changed the wording accordingly and also added a paragraph in the discussion (page 8, lines 382ff). We are currently working on the theory, which however will take much more time than can be envisaged during the revision process and may also require a different target audience/journal. Based on our analyses, we do have empirical evidence that the method works (see also response to Rev #2, comment 1; further countries in new Supplementary Figure 3) and thus, we currently view the proposed method as empirical evidence that for SARS-CoV-2 an evolutionary signal exists, from which the incidence trajectory can be deduced. Unarguably, over the next month and years, we will further evaluate the theory, improve and automate the pipelines, and assess if the method works for other respiratory infections. We have added corresponding passages to the discussion explaining the scope of the results, limitations and the outlook on page 9.

We also evaluated the method in scenarios with positive/negative selection (Supplementary Note 1, section SN.1.14, see also Reviewer # 2, comment 2). Here also, the method works well.

2. The exact interpretation of "number of mutant sequences" (m) and "number of haplotypes" (h) is unclear. Since the authors' application of the model to incidence does not seem to focus on any particular mutation (relative to a reference genome), does m represent the absolute number of sampled infections (irrespective of sequence) in a given time period (indexed by t)? In other words, does m adjust for sampling effort?

-> Absolutely, yes: If the population diverged sufficiently from the reference, then m denotes the number of sampled infections within a sequence set (= number of sequences). We believe that m therefore adjusts for the sampling effort. 'Number of haplotypes' refers to the number of unique sequences in a sequence set, as stated in the revised manuscript on page 2/3, paragraph 'Incidence reconstruction'.

3. Does h represent the number of unique genome sequences? If so, how do the authors deal with ambiguous or base calls or incomplete sequence coverage? Figure 1 does help visually explain h and m to some extent, but there needs to be a clearer explanation and rationale integrated into the main text. The term "number of mutant sequences" is particularly confusing.

-> We rephrased "number of mutant sequences", page 2/3, paragraph 'Incidence reconstruction'.

If there is incomplete coverage (in more than 10% of the sequence), the sequence will not be aligned by minimap (page 12, line 553). Our pipeline is currently entirely based on point mutations, i.e. InDels are ignored in the current version of the pipeline (missing data below the 10%). Ambiguous bases are:

(i) either treated as the reference base, if the ambiguous code contains the reference base (e.g. 'R' would be replaced by 'A', if 'A' is the reference), or

(ii) or the ambiguous does not contain the reference, a random non-ambiguous base is chosen from the set that defines the ambiguous code.

We added this information to the Methods section (page 12, paragraph 'Data and data pre-processing') and apologize that it was not provided in the initial version of the manuscript.

-> In the revision process, we also realized that the alignment filter (10% mismatch criteria in minimap) may have falsely excluded some sequences in the simulation studies (particularly during the simulation of introductions). We have adjusted the length of the sequences, as well as the

difference of the introduced sequences to the founder sequence to avoid sequence exclusion. Interestingly, removing this bug further improved the performance of GlnPipe (Fig. 1D-F and Supplementary Note 1).

4. In addition, their analogy appears to interpret the uninfected susceptible population as wild-type alleles in a population of constant size. The population dynamics of SARS-CoV-2 does not resemble a selective sweep. What are the consequences of non-sigmoidal dynamics in the number of infections? Is the origination of the mutant allele in a new haplotype analogous to the importation of SARS-CoV-2 from an external source to the population, and does this stipulate that the imported infection carries a unique genome sequence? The mutation rate is incorporated into the derivation of Khatri and Burt's result, but it does not make sense if origination corresponds instead to an importation process. I had expected to see these issues addressed substantially in the Discussion, but most of this section was used to review study results rather than discussing the model assumptions and limitations.

-> As stated in the response to comment 1, the analogy with Khatri & Burt may not be as strong as perceived by the reviewer. We apologize if our wording may have given the wrong impression.

-> Non-sinusoidal dynamics: We have tested non-sinusoidal dynamics in the extended Supplementary Note 1, section SN.1.9 (constant effective population size, time-dependent sampling) and in section SN.1.10 (step functions for the effective population size; [= extreme, non-smooth dynamics]). The method appears to cope well.

-> Importation vs. origination: In our simulations, origination and importation are different. Origination depends on the effective population size, importation not. In the extreme scenarios (Supplementary Note 1, section SN.1.8, Figure SN.10 therein) we assessed situations, where importation does not contribute to the effective population size ('stopped at the border'), i.e. does not evolve after being imported. In other words, importation was a source of 'noise' in those simulations. In some extreme cases, this may bias our inference. Thus, there is some empirical evidence that importation is not a driving factor of the proposed method: If sufficient evolution occurs, we observe that the method works: Hence, as long as the "evolutionary signal" is sufficient, the method seems to be able to reconstruct incidence profiles.

-> The mutation rate is not explicitly considered in our equation, yet. As mentioned in response to comment 1, we are currently working out the theory. When we are able to derive an explicit formula that represents the relationship between the mutation rate and ϕ , we may also be able to quantify *absolute* incidences (unlike, as currently, *relative changes*), which of course is a very high priority for us.

-> We have added a methodological discussion that also refers to the analyses performed in the Supplementary Notes SN.1.8-15 (see also comment 1.), page 8/9 in the Discussion.

5. The manuscript presents some simulation results, which is a necessary step for validating a new method, since the ground truth is known without error. Population dynamics were simulated by drawing from a Poisson distribution centred on the population size at the previous time point with a deterministic sinusoidal coefficient driving variation over time, instead of a more parametric model (such as an epochal SEIR model). I am somewhat concerned that the authors were not sufficiently critical of their model with respect to its sensitivity to incomplete sampling and importation of cases. For example, incomplete sampling was assessed by censoring infections completely at random (or stratified by time window), but systematic associations between variation in sampling rates and genomic

variation (for example, concentrated sampling of a particular district or subpopulation) may induce a more serious bias.

-> These are very good (and challenging) comments.

-> Choice of model for simulations: For the purpose of analysis, we chose the minimal modelling approach. Essentially, for model simulations, the only interesting quantity in our context is the number of infectious individuals (the 'I' in the SEIR). Moreover, we believe that an SEIR is probably the wrong modelling approach, given that SARS-CoV-2 variants (beta, gamma, delta, lambda, ...) seem to arise that are able to re-infect individuals (PMID: 33515491,33293339; i.e., the 'R' in the SEIR may be incorrect for modelling *long-term* dynamics). A suitable mechanistic modelling approach would therefore (i) either have to consider the emergence of variants by explicitly considering phenotypes (ability of strain j to infect individuals recovered from infection with strain i) in a multi-variant SEIR model, or (ii) assume that sufficient susceptibles are available at all times (SIS-like model). For the first (i) approach, many assumptions have to be made and even more parameters to be justified. The second approach (ii) may reduce to sampling from a Poisson distribution with time dependent mean (depending on I(t) and some time dependent infection rate constant $p(t) = b(t) * S$), whenever the proportion of infected individuals remains in the lower single-digit percent values (consequently $S(t) \approx S$). The latter is a very reasonable assumption for SARS-CoV-2, given that the duration of infection is short (e.g. 'prevalence estimator' function in <https://covidstrategycalculator.github.io/>).

-> The reviewer is absolutely right that severely biased sampling, e.g. only of very related cases, may induce a more serious bias in GlnPipe. In essence, if the sampling (and thus the 'evolutionary signal') is severely distorted, the method, naturally, cannot work. However, this limitation with regards to biased sampling applies to all methods available (serology, wastewater analysis, phylodynamics, diagnostics). For this reason, we have made the experience that nationally (Germany) and with our international partners (e.g. Insa-Cog, Africa-CDC, WHO), great effort is put into building up genomic surveillance networks that produce representative data.

We have performed the corresponding simulations in Supplementary Note 1, section SN.1.9 and also included this limitation in the discussion (page 8/9). In ongoing work, we are developing methods to detect and set apart such distorted signals and apply automatic filters in the pipeline. With regards to real data, we added further countries, some with very low sequencing capacities in Supplementary Fig. 3 (see also response to reviewer #2, comment 1).

-> We also want to assure the reviewer that we were VERY critically evaluating the method over the last year (note the method was already up and running at the COVID-19 Dynamics and Evolution Conference in Oct. 2020).

6. Additionally, the impact of importation on model estimates was simulated by adding genomes in which 10% of sites were randomly mutated with respect to the "founder sequence of the local outbreak". This is an excessive amount of mutation. It is not apparent to me whether this simulation setting is meant to be induce a large effect, i.e., make a conservative assessment on sensitivity of the method to importation.

-> Yes, the set-up was meant to induce a large effect to test the methods' limits. We found that the method is quite insensitive to what the imported sequences look like.

-> However, we realized that the original set-up may have introduced a bias by falsely disregarding sequences (see response to comment 3). We altered the settings accordingly to remove this bug, which made GlnPipe's reconstructions even better.

7. Lastly, percent deviation from linearity (supplementary figures) is difficult to interpret as a quantitative outcome of the simulation experiments.

-> We apologize for this inconvenience. This measure arose from the fact that, while we know that the effective population size and our estimate φ correlate linearly, we do not know the slope (and thus the *true* relation) *a priori*. Hence, the best we can do is to infer a slope and to compute the deviation from that slope. However, we also added a scatter plot with regards to the *true* and *inferred* Re in the revised Fig. 1f, as well as a contingency table and an accuracy estimate on the categorical data (see also Rev # 3, comment 9).

8. Having raised these issues, I am nonetheless quite impressed with the method presented in the manuscript - it seems to work surprisingly well on my test data. I think this will be an important contribution not only to the field of molecular epidemiology, but also for public health applications of sequence analysis. It might be helpful to quantify how much more we learn about the number of unsampled infections from this sequence analysis in comparison to conventional data sources such as test positivity rates, if possible.

-> Thank you very much for your feedback and enthusiasm, which we share 100%.

Running the program:

- I. I was able to install and run demo code on both macOS Catalina 10.15.7 and Ubuntu 18.04.5. However, I ran into problems when attempting to run GInPipe on a custom data set comprising about 5,000 SARS-CoV-2 genome sequences. First, the snakemake workflow had problems dealing with a relative path to the reference FASTA file in the configuration YAML (the program threw the following exception: "MissingIndexException: Missing input files for rule minimap_index_ref"). I had to move the YAML file into the nested directory and use the filename without any relative path prefix. This input specification needs to be more flexible.

-> We apologize for this inconvenience. In the README it says "*The specified paths in the config file should either be absolute, or relative to the work environment specified with -d in the snakemake call (see below in Execution).*" We have added more information about running the pipeline directly to the github repo.

- II. Next, I ran into a ValueError exception with the error message "invalid contig" when running the pipeline with NC_045512 as the reference genome. Replacing the sequence header with ">ref" seems to fix this problem, but there is no such requirement specified in the documentation. Overall, I found the pipeline to be quite unforgiving about path specifications and the locations of input files.

-> We tried to reproduce this error. Using arbitrary names, including "NC_045512" works, but apparently this error occurs when there are whitespaces in the header. This is also the case for the downloaded fasta file for "NC_045512" from NCBI, which might have caused the crash if the reviewer used it with the full header name. We solved this issue now: the pipeline will only take the reference name up until first whitespace. Also a suggestion to only use reference names with no whitespaces was added to the README.

- III. The third exception I encountered was associated with "rule run_binning", with "CalledProcessError in line 145". This seems to be associated with an error in the Python

script "sam_to_bins_modular.py" on line 260 with "KeyError: 0". At first, I suspected that this was due to one or more incomplete sample collection dates in the inputs. However, I found no such instance when grep'ing the input files. I then realized that the problem was that I had included spaces around the pipe ('|') delimiter between sample name and collection date fields. My reason for doing so was because the README documentation actually includes a space between "some_name" and the pipe character. Removing the excess spaces resolved this issue. Hence, the documentation needs to be more explicit about how sequence headers in the sample FASTA input should be formatted.

-> The description in the README has been changed (by error there was a whitespace after the |; apologies!). The utilized format of the header is adapted to the GISAID format (hence allows application to data downloaded directly from there).

-> We also deposited a utility function that merges meta-data files with the sequence files `add_date_from_metadata.py` in `scripts/utls` folder of the main GitHub repository.

- IV. Afterwards, I was able to run the pipeline to completion on these data. The locations of peaks in the incidence correlate plot was generally consistent with the first and second waves (with respect to daily numbers of confirmed cases) for the region represented in the data. Since these trends are fairly correlated with sample collection dates (i.e., more samples collected during waves), I also re-ran the analysis with a random permutation of collection dates among sequences to confirm that the same incidence correlate trend could not be recovered. I didn't have time to run more extensive tests.

-> This is a very nice test. Thank you so much for taking the time and for your interest in this work!

V. Source code:

- I appreciate that the authors have released their source code into the public domain under a permissive license (GPLv3). The Python code looks fairly PEP8 compliant.

- Some of the Python scripts are rather unstructured, in that the code is seldom modularized into functions, e.g., `fix_cigars_subprocess.py`. This makes it somewhat more difficult to interpret the code, and prevents users from adapting the functionality of GlnPipe into other workflows in a modular fashion. (Same goes for the R scripts - could the developers please consider turning these scripts into a package?)

-> We have considerably restructured the code and turned the respective scripts into packages.

- Some of the code style is unconventional. For example, the developers make frequent use of string concatenation instead of Python's built-in methods for formatted strings, such as `str.format()` or C-style formatted strings (with '%' placeholders).

-> Thank you very much, we have changed everything to the C-style as requested.

- External programs are being run through the shell, which is generally considered bad practice. For example, a user might be exposed to a shell injection attack if they ran a YAML configuration file with malicious text passed to `snakemake` parameters. Recommended method is `subprocess.check_call()`.

-> We are not entirely sure whether we understood the remark correctly.

If the reviewer is referring to Snakemake, we have screened the Snakemake documentation and went through various forums. We did not find any discussion that this is bad practice.

If the reviewer is referring to Python calling external programs (e.g. SAM tools), the issue has been largely resolved (however, `subprocess.check_call()` is still used once for indexing bam files (in `erase_empty_bins.py` line 45)).

- please consider using temporary files via Python module `tempfile` rather than writing to hard-coded file names like `'list_of_files.tsv'`.

- This has been changed. The script does not write files with intermediate outputs anymore.
Thanks for the remark!

- clearing the user's workspace with an `'rm()'` command in the R script is not really user friendly, particularly if a user sources one of these files in an interactive R session.

- This has been changed. Thanks for the remark!

- many functions in the R scripts need documentation; code style is a bit inconsistent, e.g., varying use of `'='` and `'<-'` assignment operations, varying use of whitespace.

- This has been changed. Thank you for the remark!

- `bam_to_fingerprints.py`, lines 103-127 would be more readable code if you used `enumerate` to iterate over `cigar`, and then unpacked the tuple into variables, i.e.,

```
for i, cigtuple in enumerate(cigar):  
    operation, length = cigtuple  
    if operation == 0: # and so on
```

- This has been changed. Thanks for the remark!

Specific comments:

- generally, the manuscript is in a very inconvenient format for review (single-spaced, narrow margins, no line numbering)

-> Apologies. Nature Comm. allows a format-independent initial submission. We realized that there were no line numbers and sent a version with line numbers to the editorial office. However, the editors were incredibly quick and the manuscript had already been sent to the reviewers. We fixed this in the revision.

- when installing GInPipe on macOS Catalina 10.15.7, I also had to install `mamba` in order for `'conda'` to detect `'snakemake'`, whereas I did not encounter this problem in Ubuntu 18.04, so this doesn't seem to be a Linux-specific issue as implied by the README document.

-> We changed the README. The recommended installation for Snakemake using `mamba` is described in the README and is not system-specific (opposite to what was stated before).

- the R package mgcv is used in `splineRoutines.R` - shouldn't this be listed as a dependency?

-> We fixed this (this was historical code which was not used anymore)

- "the sequences are placed into temporal bins b " - this is awkward phrasing, are these bins indexed by variable b , or is b the total number of bins?

-> Apologies. It is indexed by b . We rephrased the sentence (page 2/3, paragraph 'Incidence reconstruction').

- p.2, please clearly define "mutant sequences" and "haplotypes" at first use

-> done (page 2/3, paragraph 'Incidence reconstruction').

- p.2 "point estimates are prone to slight underestimation" Please provide quantitative results instead of a qualitative summary.

-> We reformulated this sentence " φ point estimates have the tendency to yield lower values" (page 3, line 118). The effect is a scaling, similar to the one shown for different filters (third last comment below "Point mutations appearing less than three times in the whole data set were filtered..."). In response to comment 7, we also provide a different assessment of the quality of incidence reconstruction (e.g. Fig.1F).

- p.3, regarding BEAST2, there are some recent advances that should enable users to run larger, low diversity (e.g., SARS-CoV-2) datasets than before, such as PIQMEE

Thank you for raising this point. Through direct correspondence with the PIQMEE developer (VB), we can say that PIQMEE may indeed be suitable for analysis of SARS-CoV-2 data sets. However, a significant increase in the number of sequences analysed (as compared to analysis using BDSKY) would only be possible if the data was sampled at very few sampling points (VB said they have tried no more than 5) and the number of unique sequences should be in the hundreds, not more. Neither is the case here. However, we have adjusted the original statement ("*However, these methods are computationally expensive, so that only moderately sized sequence sets can be used, and advanced knowledge is required to apply them properly to larger data sets.*") to better reflect the possible applications of phylogenetic methods to large data sets (page 4, section Method validation: phylodynamics).

- Figure 1A, y-axis label - why not just say "cumulative number of sequences" instead of using a formula that may frustrate some readers?

-> Thank you, we changed this.

- the variant filtering step did not seem to exclude any sequences for either the demonstration data or my own data.

-> This filtering excludes specific mutations at a specific site and not sequences (page 12, Methods section, paragraph 'Data and data pre-processing'). We also checked and corrected the README on github.

- page 4, " $R_e(\tau)$ estimates for Scotland agree almost exactly" Please provide quantitative results.

-> Specified: "The GlnPipe estimate is within 20% of the BEAST2 estimate" (page 4, line 164)

- Figure 3, since incidence estimates ϕ are correlated, the relation between the two scales (ϕ and reported cases) is arbitrary. How did you decide on a proportionality constant for drawing data on these two scales?

-> We used the respective min/max values on both axes.

- Figure 4, space permitting, it would be helpful to directly label the vertical dashed lines that correspond to different policy changes.

-> We had done this in a pre-submission version of the manuscript. However, the figures became very overloaded and hence we decided to put the explanations in the caption.

- pages 6-7, much of the text here is essentially describing features of Figure 4; I think this word count would be better invested in describing and discussing the underlying method (i.e., adapting Khatri and Burt's method).

-> We have extended the discussion of the method (page 8/9). We found that the underdetection issue is a very nice feature of the method, worth discussing with the presented examples and an important addition to the portfolio of tools to monitor SARS-CoV-2 (and possibly other respiratory infections) in the future.

- page 8, "the vast majority of reconstructed sequence data has been made broadly available through public databases" Unfortunately this is only true for a minority of countries such as Denmark and the UK.

-> Meanwhile, many national genomic surveillance initiatives make their data available, with about 2.3 million sequences on GISAID to date (including Germany; which we are making available, ever since the data was systematically collected, and against all resistances from data protection officers, after lengthy discussions with Peter Bogner and the like ... ;)). We changed "vast majority" to "many" (page 8, line 344).

- page 9, "The power of GlnPipe lies in the swift reconstruction [...] without requiring [...] masking of problematic sites in the virus genomes." This is not a computationally expensive step and benefits from domain expertise, so why not make use of this filtering step in pre-processing?

-> The pipeline does not seem to require masking, which is great in terms of reducing manual adjustments by users (which may also take time to perceive and to perform). However, we have added a utility that can mask particular sites (replace them by the reference residue).

- page 9, "The execution time appears to scale linearly with the number of sequences to be analyzed"

It would be appropriate to provide some actual results here in supplementary material.

-> We included a plot with the runtimes for some analysed countries in the new Supplementary Figure 4 and added a cross reference in the main text (page 10, line 466).

- page 10, "Point mutations appearing less than three times in the whole data set were filtered out, as they may occur due to sequencing errors." This is a problematic assumption. Depending on the size of the data set, a large number of biologically real mutations will fall below this frequency threshold. How sensitive are the results to relaxing this threshold?

-> We made this filter an optional input by the user. We found that applying this filter has a scaling effect (changing the slope of the linear correlation).

- page 10, "we deduced the nucleotide substitutions for each sequence" - so this method excludes indel polymorphisms? Is this justifiable?

-> Our current version of the pipeline focuses only on point mutations (substitutions), comment 3. Substitutions denote frequent mutational events that apparently comprise a sufficient evolutionary signal for incidence reconstruction. InDels however, are less frequent and may imply more severe phenotypic changes. We therefore suspect that Indels negatively affect signal-to-noise.

- page 11, what convolution filter, exactly?

-> We applied a smoothing filter (moving average; R routine 'filter' with window size 7, 2-sided). We added the information (page 13, paragraph 'Reconstructing the incidence history').

Reviewer #2 (Remarks to the Author):

The authors propose a novel method (GInPipe) to estimate the true incidence of SARS-CoV-2 using time-stamped viral genomic data. By analyzing the number and frequency of sequence variants at a given time, they are able to estimate the effective reproductive number and the relative incidence of infection. They validated this method using in silico data, simulating various scenarios including missing/incomplete genomic data, and the introduction of new variants into the population. Subsequently, they validated their model against real-world data from 4 countries: Denmark, Scotland, Switzerland and the Australian state of Victoria. They compared the estimates for R_e from BEAST versus GInPipe as well as relative incidence versus the actual number of reported cases in each country/region.

Overall, the manuscript is well written and represents a comprehensive validation of a complementary method to estimate COVID-19 disease incidence. This method will be especially useful when more sensitive diagnostic tests are inadequate relative to the extent of the outbreak. However, it does require the availability of a significant amount of genomic data, which is usually only available in countries with sufficient resources for PCR and sequencing. That said, there are important observations that can be inferred from their analysis - when the availability of PCR testing is reduced because of a perceived reduction in the number of cases, the genomic data from those cases may reveal more widespread, cryptic transmission; and while there is utility of rapid antigen

testing, widespread use of this less sensitive method may underestimate the true incidence of disease as indicated by genomic data.

1. It is not clear why the 4 datasets (Denmark, Scotland, Switzerland, and Victoria) were chosen. The a priori rationale for choosing these datasets needs to be stated and justified. This is important for the real-world validity of their results.

-> The choice of these countries and regions was to some extent random, but we also did have the following thought in mind: For most of the countries we expected that the pandemic was reasonably tracked by the reported cases, to ensure that the comparison between reported cases and estimated incidence is meaningful. Moreover, the following consideration were made

- Denmark: Many sequences
- Scotland: Many sequences, ?underdetection at beginning?
- Victoria: small setting; very few infections, different dynamics/waves at different times in comparison to Europe; good sequencing coverage
- Switzerland: exploratory.

-> We added a few more countries in Supplementary Figure 3 and added the corresponding text in the Results (page 6, line 240ff), as well as the Discussion (page 10).

- Japan (exploratory: ... olympic games)
- India (exploratory: probably unmitigated spread, very few sequences in comparison to number infected, emergence of delta variant)
- Chile (exploratory: very few sequences; high rates of vaccination)
- South Africa (exploratory: potentially a lot of unnoticed spread, fewer sequences in comparison to number infected, emergence of beta variant)

2. The mutation rate is not constant throughout the SARS-CoV-2 genome. There are regions under neutral pressure whereas other regions are under selective pressure. In addition, there are synonymous and non-synonymous mutations. Could the method be improved by using only neutral regions of the genome and/or non-synonymous mutations?

-> Practically, whether positions are neutral may not be known *a priori*, as even synonymous mutations may be selected (non-coding RNA, or codon bias). However, we included a utility in GlnPipe that allows us to mask particular sites (see also Rev. # 1, fifth last minor comment).

-> We set up a simulation example (Supplementary Note 1, section SN.1.14), where we also incorporated sites under selective pressure. We chose a parameter setting, such that the average fitness of the viral population would increase up to factor 2 during simulations (Fig. SN.22 in Suppl. Note SN1). Note that the indicated fitness value is similar to the putative fitness advantage of the delta variant over the wild type of approx. factor 2). In summary, the method still works well when some sites are under selection.

Could the authors explain the rationale for grouping by Pango lineage and subsampling within lineages?

-> The reason for subsampling within Pango lineages for the phylodynamic analysis, as only done for the D.2 lineage in Victoria, was the very high proportion of sequences assigned to D.2 in the data set and the relatively low subsampling percentage. Taken together, this would have led to the loss of most of the non-D.2 sequences, especially those comprising the background during the D.2 outbreak. To account for this non-random subsampling, we model and estimate a separate sampling proportion for lineage D.2 compared to the non-D.2 lineages.

In general, we subsample the full data sets randomly through time to decrease the total number of sequences to a point at which they can be analysed in a reasonable time with the phylodynamic method used. We then group the sequences in the subsampled data set by Pango lineage in order to roughly approximate independent introductions into the area, such that most transmission events in the trees happened inside of the considered area. Even though the Pango lineages do not provide an exhaustive separation of intra-area clusters, they are defined in a way that new emerging clades within the global SARS-CoV-2 phylogeny are identified, especially when spreading into a new region. We therefore assume that, although we cannot identify all introductions, we are able to separate clusters of potentially different dynamics using this clustering method. We have clarified this in the revised manuscript.

Reviewer #3 (Remarks to the Author):

The aims of this paper – to approximate incidence using genetic data alone and to compute changes in the probability of reporting are both important and interesting. Characterising the incidence of cases and even deaths is not simple, especially in the face of detection delays and under-ascertainment. An approach that can circumvent some of these problems would be a valuable addition to the outbreak response toolkit. This paper makes some good progress towards these aims but I have several major concerns around validation and accuracy, which need to be resolved for this analysis/methodology to be convincing.

-> We thank the reviewer for their appreciation of our work.

1. The validation on simulated data is not yet sufficient. This is especially important for a paper

proposing a new method. A couple more examples with different dynamics should be included and then some statistics computed to showcase accuracy (e.g., considering the lag and scaling between the true incidence and inferred correlate). In particular, the current example shows clear differences ($t = 30-50$ and $t > 100$) that need to be explained and accounted for before the claim of accuracy can be upheld.

-> In Figure 1D, the apparent difference ('lag') may have been visually deceiving due to the scaling of the respective y-axes. We reran the simulations for additional accuracy analyses, this time with longer sequences but same settings. The re-running was necessary, since we found a small bug in the code (mapping filter in minimap), as outlined in response to Rev #1, comment 3. The resulting new Figure 1D-F does not show this "lag". Other analyses in Supplementary Note 1 did NOT point towards systematic 'lags' in GInPipe's estimates (more below). We also added a more qualitative analysis of apparent differences between simulations and incidence reconstructions in Fig. 1F, see also comment 9.

-> We replaced 'accurately' in the manuscript, since accuracy may depend on the goal of the analysis. While we can compute correlation coefficients, we do not know the precise scaling factor between our incidence correlates and the true incidence, hence, it is currently difficult to quantify if, and how much the prediction is off in particular scenarios *quantitatively*. With regards to *qualitative* comparisons we added analyses (below and comment 9).

-> More examples and analysis:

- a) We included a few additional countries (Supplementary Figure 3; see also response to Rev #2, comment 1), some with rather low sequencing coverage (Chile, India, South Africa). Generally, also based on the additional analyses, we would consider the method to perform well with real data.
- b) In terms of simulations, we performed additional tests, e.g. evaluating whether
 - i) a time-varying, drastic change in the sampling proportion (= sequencing coverage) has effects, Supplementary Note 1, section SN.1.8.

We found that the sampling proportion does not affect the incidence reconstruction

- ii) Lag-time: We assessed whether GInPipe can reconstruct non-smooth pandemic dynamics (sudden increases or decreases of the number of infected individuals by several factors), Supplementary Note 1, section SN.1.13.

We found that if the pandemic dynamics are too extreme (step function), a 'smearing out' may appear. This was however only observed for drastic increases of the number of infected individuals (Supplementary Note 1, Figure SN.21 therein). However, the tested step functions are likely more extreme than real pandemic curves.

1) Lag times in φ in relation to *increasing population sizes during simulations* can occur when mutation rates are very low (in comparison to the population dynamics). In all applications of the method to real data, we do not observe this type of delay. I.e., φ typically increases before, or coinciding with increases in reported cases. Thus, we are confident that this lag does not occur for SARS-CoV-2.

2) A lag time with regards to *decreasing population sizes during simulations* can arise when the variety of haplotypes persists despite decreasing population sizes. Therefore, this lag arises when the 'renewal rate' is low (rate to become noninfectious). The rate to become non-infectious is however large in SARS-CoV-2 (see also comment 5), such that we anticipate

to observe a small 'lag effect' with SARS-CoV-2. We speculate that the 'apparent lags' when comparing to case reporting date (Fig 3), may actually be a result of the diagnostic behaviour, i.e. underreporting of cases after the peaks when people are 'tired of the pandemic'. For the third wave in Scotland we also performed additional analyses with EpiInf (Supplementary Figure 1; see also comment 4), which also predicted a more long-lasting third wave.

- iii) We also tested whether *selection* affects GInPipe, Supplementary Note 1, section SN.1.14.

We did not find major effects of selection on GInPipe.

- iv) We tested whether biased sampling affects GInPipe, Supplementary Note 1, section SN.1.9.

We essentially found that when closely related sequences are more likely to be sampled, we see no systematic effects on GInPipe. However, if the sampling of diversity changes over time (e.g. a switch from 'random' to 'genetically similar' sampling) the evolutionary signal becomes temporally distorted and incidence reconstructions are affected. We added some more remarks regarding this last point to the manuscript (see also Rev #1, comment 5).

2. The approach to simulated epidemics also seems somewhat strange (especially given the use of the Wallinga-Teunis method later). Why not use a renewal model to more accurately simulate what an epidemic might look like (and which is the model behind the Wallinga-Teunis)? The key difference from the current approach would be the use of a generation time distribution (which is better suited for properly considering incidence on daily scales as the paper provides) rather than a simple branching process with fixed generations.

-> We used the simulations only to generate data on which we can evaluate the GInPipe method in silico (Supplementary Note 1). As pointed out by the reviewer, we used a minimalistic modelling approach for these simulation studies (see also Reviewer #1, comment 5). We chose the method (discrete time) because it is computationally efficient. Essentially, for the simulation studies in Supplementary Note 1, 'time' (whether continuous on a real-, or virtual scale, or discrete) is not relevant. To illustrate this argument, we also performed simulations using exponentially distributed generation times (classical Markov Jump Process formalism) in Supplementary Note 1, section SN.1.10. Sampling from more complex generation times is also possible (e.g. PMID: 33899035, Fig. 1C therein, for example using the EXTRANDE algorithm), but will not affect any of the conclusions made in Supplementary Note 1, other than making the simulations more time-consuming. So, in essence, we are convinced that the simulation method is well-suited for the purpose of testing GInPipe on simulated data.

3. The comparisons of R_e via BDSky and the Wallinga-Teunis approach do not seem that consistent – more analysis is needed, and the confidence intervals of both approaches do not seem that clear. While the need for piecewise constant R_e from BDSky is understandable, there still are discrepancies that warrant a closer look.

-> We agree with the reviewer that the comparison of R_e from BDSky vs. GInPipe is challenging. We, however, deliberately wanted to include the comparison with an entirely different method (BEAST) that utilizes the same data. Additionally, we are comparing independent measures (here R_e) that, no matter the method used to infer them, should agree with one another. There are obvious differences in the methods such as a) the need for a piecewise constant R_e in BDSKY vs. the

ability to derive a continuous function in GInPipe. The largest source of discrepancy particularly early in the pandemic is our relatively crude clustering approach, please see also the answer to the next comment. For better comparability, we have now used our piecewise constant results from BDSKY to infer continuous incidence trajectories using the BEAST2 package EpiInf for Scotland (Supplementary Figure 1).

4. Why not also compare the Ne with coalescent approaches? It does not appear the Ne from the method chosen has been considered against more standard approaches such as <https://academic.oup.com/mbe/article/22/5/1185/1066885>. It would be good to know if the correlation between Ne and incidence is general.

Thank you for the suggestion. Many tree-based phylodynamic methods such as the one Bayesian coalescent skyline plot (BSP) suggested by the reviewer make the assumption that the phylogeny is a good approximation of the transmission tree. This is one of the reasons for the necessity to split the large country-wise data sets into clusters. In contrast to the BDSKY approach used here, the full data set cannot be analysed by joining all clusters (approximating independent introductions) into a single analysis, because the tree intervals cannot easily be adjusted to certain points in time to ensure the temporal alignment of separate trees. Thus, all sequences have to be analysed in one tree. This leads to the reconstruction of a large number of coalescent events outside the considered region, likely biasing the estimate of the effective population size over time (see also comment 11). We have nevertheless tried running BSP on the full trees, however, the method does not converge properly. This is most likely due to the large number of sequences, requiring the reconstruction of trees with over 2,000 tips.

However, to provide a better comparison to the GInPipe incidence estimates, we have set up the BEAST2 package EpiInf (PMID: 31058982) to infer incidence trajectories over time from the results obtained from the presented BDSKY analyses (see point 3). As stated in the manuscript, setting up and performing phylodynamic analysis requires considerable expertise, fine-tuning and computational time to derive meaningful results, which altogether demanded the majority of time during revision. Therefore, we only did this for one country. The resulting incidence estimates from EpiInf for Scotland are shown in Supplementary Fig. 1. The comparison shows that EpiInf, GInPipe and reported cases agree overall. The EpiInf estimation for the epidemic waves one (April '20) seems to lag slightly behind, and wave two (November '20) is slightly underestimated in comparison with GInPipe and reported cases. Both EpiInf and GInPipe hint towards a longer lasting third wave (Jan '21) in Scotland. We also see an epidemic wave in August '20 for EpiInf that is not supported by the reporting data or using GInPipe. We suppose that this may be an artefact that is caused by the crude clustering of the sequences in phylodynamic analysis. As mentioned in the manuscript, the phylodynamic inference is very sensitive to clustering and it may not be possible to find a clustering setting that produces robust results for all different countries that were analysed in the manuscript (we chose one clustering approach for all BEAST analyses). We have revised the manuscript accordingly (Results: page 5, line 188 and Methods: page 14, section 'Phylodynamic analyses')

5. The methods of <https://royalsocietypublishing.org/doi/full/10.1098/rstb.2010.0060> have explicitly investigated relationships among Ne and prevalence/incidence. I think this paper should comment on those links since it proposes another correlation.

-> If we understand the reviewer and the mentioned paper correctly, it is stated

- a) that coalescent times (and generation times * Ne) may correlate with incidence, but not with prevalence, which may be out of phase or temporally shifted.

For SARS-CoV-2, the duration of infection is typically short, such that the temporal shift is very small (50% of infections are cleared a week after symptom onset (~diagnosis & sampling time), PMID: 33899034). However, we noted that ‘infected individuals’ is sometimes used ambiguously (i.e. referring to the cumulative number of individuals that have been infected). We therefore revised the manuscript and refer to either incidence or *actively* infected individuals.

- b) Secondly *“In this model, as the time between infections changes, the use of a single transformation of time to fit the early stages of the epidemic results in an overestimation of the true number of infected individuals in the later stages.”* . Please refer to the answer to the next comment.

6. Some studies also noted that the generation time is effectively the time between infections (Pomeroy et al. 2008; van Ballegooijen et al. 2009), and not the duration of infectiousness, but did not recognize that this changes throughout an epidemic. Hence, a single transformation of time, which is commonly used to estimate N_e from temporally sampled sequence data, cannot be used to recover the ‘effective number of infected individuals’

-> We thank the reviewer for this statement and hope that we understood the comment by the reviewer correctly. Regarding GInPipe and following the discussion in the paper (PMID: 19910379), we agree with statements made therein. [For hepatitis B:] *“A reduction in genetic diversity can be due to a decline in the number of infections but also to a shorter generation time or an increase in the average and variance of the number of secondary infections produced by one infective individual”* .

Shorter generation times: the statement above refers to an infection (Hepatitis B), which can be *onwards* transmitted either within weeks, or years after infection, i.e. there is an immense range of evolutionary time and selective pressure that shapes the viral quasispecies, before onwards transmission. Therefore, if the generation time for HBV shortens considerably, for example to weeks, the “evolutionary signal” would also be completely altered, as stated in PMID: 19910379. I.e., a variant that is onwards transmitted at later time points after infection may have diversified considerably from the founder virus, whereas an early transmitted variant may not.

This is entirely different in SARS-CoV-2, where any within-host quasi-species dynamics have comparatively little impact on the transmitted variant (this is why SARS-CoV-2 mutates so little at the population level, compared to the fidelity of the RdRp; A similar observation has been made also for other respiratory infections, e.g. paramyxoviridae in PMID: 18217182). I.e., the virus may *usually* be transmitted before quasi-species break through, and the transmitted virus may *usually* be a result of (almost) clonal expansion of the founder virus, or a founder virus that acquired mutations during the first replication cycles. Because the transmitted virus usually already has very few (if at all) mutational differences with regards to the founder virus, further shortening of the generation time does not significantly impact on the genetics of the transmitted variant.

We are aware of these differences between the distinct viruses (e.g. respiratory vs. sexually transmitted), and clearly state that we believe that GInPipe only works for infections that are passed on within a very short time after infection on page 8. We also added a comment about super-spreaders on page 9 in the discussion.

7. In the supplement the importance of binning strategies is noted. Can some comment in the main text be given for what selection approach was taken? Is there some good theoretical reason? The bias-variance trade-off of bins is well known at least for N_e <https://academic.oup.com/sysbio/article-abstract/68/5/730/5307781>. Can some related comment be made in the choices of this approach?

-> Yes, this was potentially hard to find in the Supplementary Note 1 (last paragraph of section SN.1.5 in the original note). Essentially, on the one hand, the bins have to be large enough in order to contain enough mutational information, but on the other hand not too large such that the time resolution is sufficient (e.g. 'peaks' and 'valleys' within the population dynamic can still be captured). We added a corresponding statement to the methods section, page 12, paragraph 'Construction of temporal sequence bins'.

In Supplementary Note 1, section SN.15 we also further observe a relation between the mutation rate (\sim evolutionary signal) and the bin size: If the mutation rate is lowered, the evolutionary signal per sequence is lowered and hence larger bins need to be chosen to contain enough 'evolutionary signal'.

8. We hypothesize that the genetic data alone holds information about the pandemic trajectory – I would remove this (as it is what makes phylodynamics as a whole useful) and go to the next line, which is the actual hypothesis specifically examined here.

-> We modified the sentence accordingly without breaking the flow of the text (page 2, lines 47-48).

The approach builds on recent work by Khatri and Burt... – could you add a line with some additional explanation here to improve readability for those unfamiliar with this paper? This is particularly helpful since this is a major point underlying the paper.

-> Khatri & Burt: We have better clarified the relation to the paper by Khatri & Burt, as outlined in response to reviewer #1, comment 1.

9. We observed a strong ($r = 0.96$)... This correlation is not as informative as it could be. A similar association but done per time point would be more useful to confirm if the seeming lag between the inferred and true N_e is upheld or an artefact. Such lags are important for a method providing incidence estimates given what of the key differences between incidence and reported cases is indeed the lag, the influence of which has been debated. E.g., see <https://journals.plos.org/ploscompbiol/article?id=10.1371/journal.pcbi.1008409>

-> We thank the reviewer for this feedback. We spend a lot of time investigating the 'seeming lag' (response to comment 1, Supplementary Note 1, section SN.1.13) and to better visualize the data. Regarding the latter, we decided to include an additional figure that compares the respective $Re(t)$ estimates (which we estimate for each time point t). In particular, in the new Fig 1F one can see both the *qualitative* and *quantitative* congruence of the simulated- vs. reconstructed dynamics, i.e. for the upper right quadrant of Fig. 1F both dynamics are increasing ($Re(t) > 1$), for the lower right they are both decreasing ($Re(t) < 1$) and the off-diagonal elements denote *qualitative* mismatches. We hope that this additional representation satisfies the reviewer.

10. Our analyses showed that the method can still accurately reconstruct incidence histories over time, when data is missing or when data sampling is unbalanced – this needs to be better explained and qualified/validated. It seems counterintuitive given that sampling is well known to be a major source of bias both in genetic data and case data (and for estimating either Re or N_e). If this claimed robustness does hold then it is worth including background for why this would be an advance/important trait of the method e.g., for case data/ Re see

<https://academic.oup.com/aje/article/178/9/1505/89262?login=true> and for genetic data/Ne
<https://academic.oup.com/mbe/article/37/8/2414/5719057?login=true>

-> Robustness to sampling Bias: We have performed quite a few additional simulations to test the effects of sampling on incidence reconstruction with GInPipe (see also response to comment 1). Basically,

- In Supplementary Note 1, section SN.1.8, we show that the sampling proportion does not introduce any biases with regards to incidence reconstruction using GInPipe. In contrast, the same experiment would introduce biases in standard phylogenetic reconstruction as shown in <https://academic.oup.com/mbe/article/37/8/2414/5719057?login=true> (pointed out by the reviewer) but can be overcome by the epoch sampling skyline plot (ESP).
- In Supplementary Figure 3, we show the incidence reconstruction for settings with much fewer viral sequences available (India, South Africa, Chile). In particular for South Africa, where the fewest sequences are available, the incidence reconstruction seems to be particularly good (In India we suspect quite large underreporting).
- A possible limitation, which we included in the discussion (page 8/9), is sampling that affects the diversity, see also reviewer #1 comment 5. The corresponding simulations were conducted in Supplementary Note 1, section SN.1.9. This type of sampling severely manipulates the input (“evolutionary”) signal, hence any method, including phylodynamic reconstruction is likely affected by this kind of manipulation.

-> Importance of the method: We envision that GInPipe could serve as a complementary tool to case reporting data, in particular when the diagnostic surveillance infrastructure may be insufficient. We stated this utility of GInPipe in the abstract.

11. Finally, we evaluated whether introductions of foreign sequences affect the reconstruction of incidence histories – this is another counterintuitive point since introductions/imports affect estimates of key epidemiological parameters as has been found across COVID-19. I think this needs more qualification and detail.

-> Exactly! This is actually a particular strength of the proposed method over phylodynamic reconstructions, which are very sensitive to introductions.

- The sensitivity of the **phylodynamic** methods with regards to introductions is caused by the attempt to coalesce the lineages. Obviously, introduced lineages would affect coalescent times and consequently estimates of epidemiological parameters derived from them (in our BEAST analyses we circumvented this issues by building separate phylogenies for the distinct lineages), see also comment 4.
- The proposed method (GInPipe) does not coalesce lineages. Frankly, it does not even consider the relatedness of lineages. In essence, introductions will simply appear as additional haplotypes. In the (quite unrealistic case) that these introduced sequences (haplotypes) do not contribute to the pandemic, and represent a considerable proportion of all haplotypes (e.g. >> 10%; Supplementary Note 1, section SN.1.12, Figure SN.19 therein) the method may overestimate incidence. However, this extreme example is there to test the limits of the method, and very unlikely to ever be encountered with real data.

12. For the second wave, reconstructed incidence histories correspond to the reported cases – this

does not seem quite right as reported cases themselves do not correspond with the incidence. Please clarify what should be comparable.

-> We reformulated the sentence. We meant to say that the *profiles* match (page 5, line 226).

13. Taken together, these lines of evidence suggest that evolutionary change of SARS-CoV-2, the effective viral population size, and the number of infected people are correlated – could some more detail and intuition be provided to help readers understand why this correlation, which is the main assumption behind the method, is valid?

-> We realized that the formulation may have been misleading and cryptic. We reformulated the corresponding paragraph and tried to explain better why we think that there is a link between the viral evolution that is observed in patient samples, and the number of infections for SARS-CoV-2. (page 8)

14. Finally, we envision that the method will be particularly useful to estimate the extent of the SARS-CoV-2 pandemic in regions where diagnostic surveillance is insufficient for monitoring, but may still yield a few samples for sequencing – has this point been demonstrated as possible?

-> Yes, thank you for raising this point. We added a few more countries (Japan, India, Chile and South Africa), some of which have little diagnostic surveillance, in Supplementary Figure 3 and added the corresponding text on page 6, starting in line 240 (see also comment 1 and Rev#2, comment 1)

15. The reproductive number $Re(t)$... was drawn from a log-normal distribution ... which is changed to $N(48.8;1)$ after first control measures are implemented in the respective area – can some more intuition and explanation be provided for these choices?

-> The reasoning behind the prior distributions that we set for the three epidemiological parameters is the following: Since we want to estimate the reproductive number, we have chosen a distribution that is rather uninformative in the range of parameters that are allowed. Therefore, we used the lognormal distribution, which does not assign any probability mass to values smaller than 0, and parameterized it with 0 and 4, yielding a prior mean for the reproductive number of 1 with a relatively high standard deviation. The become-uninfectious rate, in contrast, we do not aim to estimate and instead constrain it strongly to account for correlations between the BDSKY parameters. For COVID-19, the end of the infectious period lies, on average, at 13.5 days post infection (PMID: 33899034). Therefore, we use a strict prior centered around 27.1 per year (corresponding to a duration of infection of 13.5 days), namely the normal distribution with mean 27.1 and standard deviation 1, in the naive population. However, in all four countries considered here, strict non-pharmaceutical interventions (NPI) were implemented in early 2020 when case numbers continued to rise. These interventions included stay-at-home orders for people with respiratory symptoms and quarantining of infected and contact individuals, which reduces the time span where individuals could infect others. Since the become-uninfectious rate in our model captures the inverse of the effective duration of infectiousness, it is not only determined by the course of the disease, i.e. recovery or death, but also by the possible changes in behaviour leading to a decreased time of infectiousness. Here, we assume that after the implementation of first NPIs, infected individuals are being diagnosed and quarantined or start to self-quarantine on average 7.5 days after being infected. This corresponds to a rate to become uninfectious of 48.8 per year.

-> As discussed in response to comment 6, GInPipe does not require these adjustments for the estimation of the incidence correlate.

Reviewers' Comments:

Reviewer #1:

Remarks to the Author:

I appreciate the additional work that the authors have put into revising their manuscript and source code. I think there was some misunderstanding about what I meant by making the code more modular (i.e., compartmentalizing blocks of code into functions), but this largely boils down to divergent coding styles. Additionally, there seems to have been confusion about population dynamics that do not correspond to the assumptions of the model. I was referring to "sigmoidal" curves, not "sinusoidal". However, this is tied up in a more direct interpretation of Khatri and Burt's model, and it seems that the analogy is meant to be looser.

I was a bit disappointed to learn that the theoretical underpinnings of the method are not well understood, making the method itself a bit of a "black box". Even so, the simulated and empirical findings are sufficiently convincing, and I look forward to seeing a more detailed investigation of the model in subsequent work.

- Abstract, missing spaces "August2021" and "205million"
- line 544, it is unusual to use uppercase in "InDels", usually this is written "indels".

AP

Reviewer #2:

Remarks to the Author:

I agree with the other reviewers that this paper describes an important and useful method which adds to the tools available for outbreak modelling when genomic-level data is available. The authors have thoughtfully addressed all my concerns.

Reviewer #3:

Remarks to the Author:

I am pleased with the depth and focus of this revision. I particularly appreciate that the lag was found to be an artefact (which was a previous major concern) and enjoyed the additional sensitivity tests that were done. Results look much more robust now. One point the authors may consider in future is in upgrading their R_e estimates from the classic WT. Recent approaches (e.g. EpiFilter, which appears to combine WT with EpiEstim) might likely smooth some of the fluctuations in R_e that they found and aid comparison against BDSky.